# From Diagrams to Code: Multilingual Programming with Visual Design

**Linzheng Chai** [1]  **Jian Yang** [1]  **Shukai Liu** [1]  **Wei Zhang** [1]  **Liran Wang** [1]  **Ke Jin** [1]  **Tao Sun** [1]  **Congnan Liu** [1]
**Chenchen Zhang** [2]  **Hualei Zhu** [1]  **Jiaheng Liu** [3]  **Xianjie Wu** [1]  **Ge Zhang** [2]  **Tianyu Liu** [2]  **Zhoujun Li** [1]

## Abstract

In modern software development, particularly in emerging "vibe coding" paradigms, implementation increasingly begins with visual interactions between users and AI coding assistants, where system architectures are communicated through visual designs before coding. This shift requires AI systems capable of interpreting diagrams and generating code across multiple programming languages, yet progress is constrained by the lack of large-scale multimodal training data and standardized evaluation benchmarks. To address this gap, we present $M^2$C-INSTRUCT, a multilingual multimodal instruction-tuning dataset with over 13.1M samples spanning 50+ programming languages, designed for visual understanding and diagram-grounded code generation. Leveraging $M^2$C-INSTRUCT, we train $M^2$-CODER, a multilingual multimodal software developer that integrates visual design inputs with textual instructions, and introduce $M^2$EVAL, a novel multilingual benchmark for evaluating multimodal code generation. Experiments show that our 7B $M^2$-CODER achieves performance comparable to much larger 70B+ models, demonstrating the quality and effectiveness of $M^2$C-INSTRUCT. Together, $M^2$C-INSTRUCT, $M^2$-CODER, and $M^2$EVAL provide essential infrastructure for visual-assisted programming in vibe-coding and visual-interactive development workflows.

## 1. Introduction

In modern software development, especially under emerging "vibe coding" and visual-interactive paradigms, implementation increasingly starts from visual artefacts rather than plain text. System architectures, workflows, and UI behaviours are first specified through diagrams, mockups, and design sketches, then translated into executable code in multiple programming languages. This raises a key question: can a model read visual design intent and implement consistent code across languages?

Large language models (LLMs) and Large Multimodal Models (LMMs) such as Claude (Anthropic, 2024) and GPT (OpenAI, 2023; 2025) achieve strong performance on many code-related tasks. Open-source code LLMs like StarCoder (Li et al., 2023) and DeepSeekCoder (Guo et al., 2024) also perform well on text-only benchmarks (Jain et al., 2024; Aider Team; Jimenez et al., 2023; Zhuo et al., 2024). However, these models are trained and evaluated almost entirely on textual inputs, implicitly assuming requirements are fully specified in text, whereas in practice visual artefacts are often the primary carrier of system design, especially in multilingual codebases.

Recent work introduces visual inputs into code LMMs, for example Design2Code (Si et al., 2025) and Web2Code (Yun et al., 2024) for UI/HTML generation, and MatPlotBench (Yang et al., 2024c) and ChartCoder (Zhao et al., 2025) for chart-to-code. These efforts focus on narrow domains, are mostly monolingual, and usually treat diagrams as auxiliary hints to a text prompt rather than the main specification. In contrast, industrial software architects frequently start from diagrammatic descriptions of class structures, design patterns, or process workflows (see Figure 1), and developers are expected to read these diagrams and implement consistent code in the target language.

A main obstacle to such visual-first, multilingual workflows is the lack of large-scale multimodal training data and standardized benchmarks that directly target the "from diagrams to code" setting. Existing multimodal code datasets are limited in scale, language coverage, or task diversity, and do not systematically evaluate models as visual software developers rather than purely text-based code generators.

To address this gap, we propose the **M**ultilingual **M**ultimodal software developer for **Code** generation ($M^2$-CODER). $M^2$-CODER consumes visual software design artefacts together with textual instructions and generates executable code across multiple programming languages. We use UML diagrams and flowcharts as "Visual Work-

[1]CCSE, Beihang University [2]M-A-P [3]Nanjing University. Correspondence to: Jian Yang <jiayang@buaa.edu.cn>.

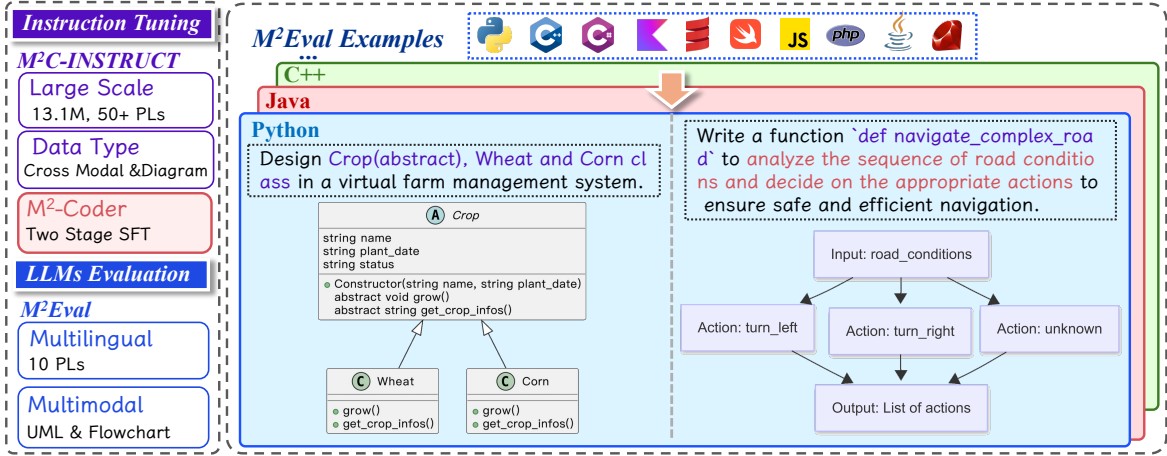

*Figure 1.* Overview of M²-CODER, M²C-INSTRUCT and M²EVAL.

flow", the dominant visual tools for representing system architecture, design patterns, and process logic before implementation. As illustrated in Figure 1, UML diagrams capture high-level structure and object relationships, while flowcharts express control flow and algorithmic logic, enabling M²-CODER to recover intended behaviour rather than rely on local code patterns.

Building such a model requires both training data and evaluation infrastructure. We construct M²C-INSTRUCT, a multilingual multimodal instruction-tuning dataset with over 13.1M samples across 50+ programming languages, including cross-modal problems (code rendered as images) and diagram-centric problems (where key information appears only in the diagram). Based on M²C-INSTRUCT, we train M²-CODER via a two-stage process: first aligning vision and language on large-scale multimodal code instructions, then fine-tuning on high-quality visual workflow problems. Finally, we introduce M²EVAL, a human-curated multilingual benchmark for diagrams-to-code generation in 10 programming languages. Our experiments on M²EVAL show that a 7B M²-CODER can match or surpass much larger 70B+ LMMs on multilingual multimodal programming tasks. Our contributions are summarized as:

- We introduce M²C-INSTRUCT, a large-scale multilingual multimodal code generation dataset with over 13.1M instruction–answer pairs across 50+ programming languages, and use it to train M²-CODER, a code generation model that leverages UML diagrams and flowcharts to better match architectural intent.

- We present M²EVAL, a human-curated benchmark for evaluating multilingual multimodal code generation. M²EVAL covers 10 programming languages and a diverse set of visual-workflow problems.

- We show that our 7B M²-CODER matches or outperforms much larger 70B+ LMMs on M²EVAL, and analyze limitations in visual understanding, instruction following, and

higher-order programming skills.

## 2. M²EVAL Benchmark

### 2.1. Task Definition

Given the $k$-th language $L_k$ from the set $L_1, ..., L_K$ of $K$ programming languages(PLs), we input the instruction $I^{L_k}$ and diagram $D$ into the LMM $\mathcal{M}$ to generate a code-related response $c^{L_k}$, sampled from the distribution $P(c^{L_k} \mid I^{L_k}, D; \mathcal{M})$. We then extract the code from $c^{L_k}$ and execute it with the corresponding test cases $u^{L_k}$ to obtain the final output. The output is expected to match the results specified by the test cases.

The process can be described as:

$$r^{L_k} = \mathbb{I}(P(c^{L_k}|I^{L_k}, D; \mathcal{M}); u^{L_k}) \qquad (1)$$

where $\mathbb{I}(\cdot)$ is the indicator function by executing the generated code with the given test cases $u^{L_k}$. when the generated code $c^{L_k}$ passes all test cases, the evaluation result $r = 1$, else $r = 0$.

### 2.2. Data Curation Process

As illustrated in Figure 2, we follow a three-step process to build high-quality M²EVAL: (1) design prototype problems in Python based on common programming concepts; (2) convert them into multimodal problems by adding diagrams and removing redundant text; and (3) translate them into multiple programming languages. A general overview is presented here, with further details in Appendix A.

**Prototype Problem Design.** We design prototype Python problems based on scenarios and varying programming knowledge levels. An LLM generates initial prompts, solutions, and test cases. These are then manually refined through multiple iterations with the LLM to ensure accuracy, alignment with design goals, and comprehensive test coverage.

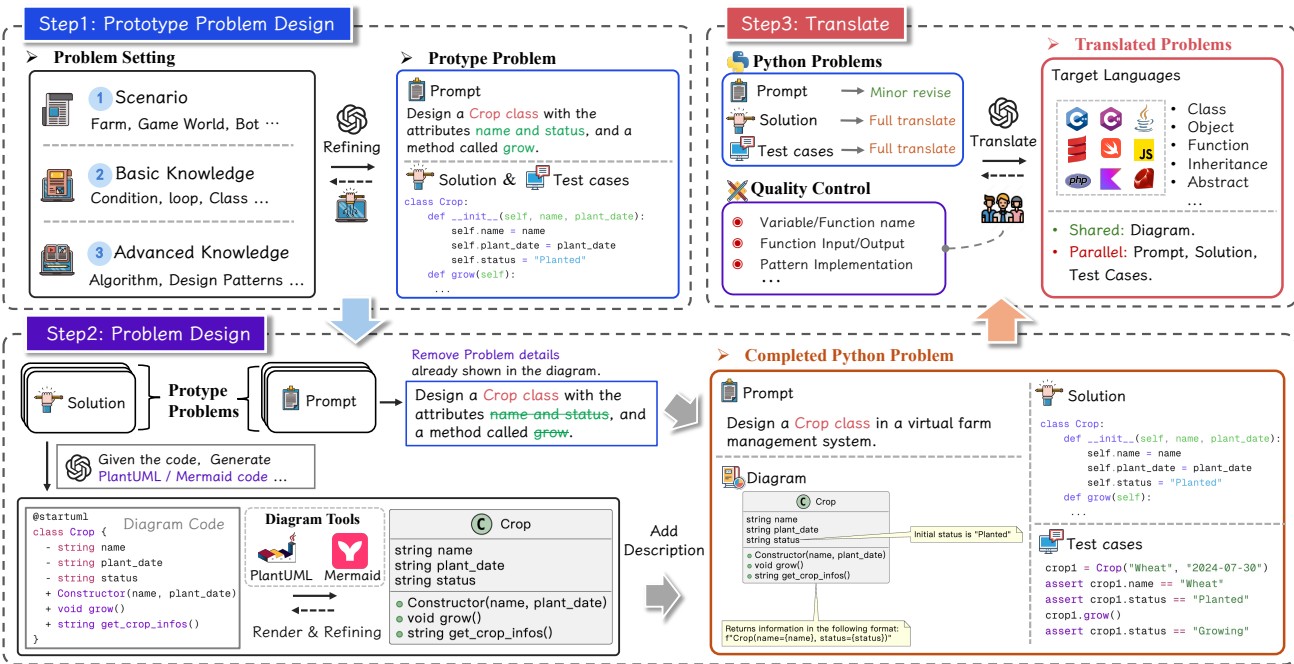

*Figure 2.* The curation process for M$^2$EVAL: (1) designing Python prototype problems grounded in core programming concepts; (2) transforming these into multimodal problems by incorporating diagrams and refining prompts; and (3) translating the problems into multiple programming languages.

**Problem Design.** Next, for each prototype problem, we generate diagrams (PlantUML/Mermaid) from the solution code using an LLM, then manually refine these diagrams for structural and semantic accuracy. We revise the problem prompts by removing information redundant with the diagrams, making the prompt alone insufficient for a correct solution. Essential details are added directly to the diagrams to ensure the problem remains solvable when both the prompt and diagram are provided. This yields Python problems with a prompt, diagram, solution, and test cases.

**Translate to other PLs.** Finally, the Python problems are translated into nine other programming languages. Prompts are adapted (e.g., by modifying function names), and solutions and test cases are fully translated. This task is completed by nine volunteers, who use LLMs for assistance and manually verify the translations for accuracy, ensuring consistency in variables and input-output alignment.

### 2.3. Data Statistics

Key statistics for the M$^2$EVAL benchmark are in Table 1. The benchmark contains 300 problems: 30 unique concepts, each in 10 Programming Languages (PLs), yielding 30 parallel instances per PL. Problem descriptions average 89 tokens (max 388), tokenized with Qwen2.5-Coder. It includes 30 distinct images, one per concept, shared across all 10 PL versions. Image dimensions range from 159-1978px (height) and 338-2081px (width). All 300 problems have

solutions, averaging 326 tokens (max 826). Each solution is evaluated against an average of 9 test cases.

In Table 3, we compare M$^2$EVAL with other multimodal code datasets. Notably, M$^2$EVAL provides a significant supplement by supporting 10 programming languages (PLs), providing 13.1M samples for instruction tuning, and introducing a novel "Visual Workflow" task type.

## 3. M$^2$C-INSTRUCT and M$^2$-CODER

### 3.1. Data Overview

As shown in Table 2 and Figure 3, M$^2$C-INSTRUCT is a comprehensive dataset of 13.1M problems divided into two stages, corresponding to the two-stage fine-tuning of M$^2$-CODER. Stage-1 contains 12.9M problems spanning over 50 programming languages, accompanied by 42.3 million images. Stage-2 features 168K problems across more than 20 programming languages, each with an associated image (168K images in total). For more detailed information on the data statistics and construction process, please refer to the Appendix B.

### 3.2. Data Construction

**Source Data Preparation.** As illustrated in Figure 4, M$^2$C-INSTRUCT comprises two stages of data preparation for M$^2$-CODER fine-tuning. In the first stage, we collected a large-scale, multilingual code dataset from GitHub. Lever-

*Table 1.* M$^2$EVAL statistics.

| Statistic | Number |
|---|---|
| Total problems | 300 |
| - PLs | 10 |
| - Problems per PLs | 30 |
| Max. length | 388 |
| Avg. length | 89 |
| Images | 30 |
| - Max. height | 1978 |
| - Min. height | 159 |
| - Max. width | 2081 |
| - Min. width | 338 |
| Solutions | 300 |
| - Avg. Test Cases | 9 |
| - Max. length | 826 |
| - Avg. length | 326 |

*Table 2.* M$^2$C-INSTRUCT statistics.

| Statistic | Number |
|---|---|
| Total problems | 13.1M |
| Stage 1 problems | 12.9M |
| - PLs | 50+ |
| - Images | 42.3M |
| - Image Height | 24-13345 |
| - Image Width | 10-57246 |
| - Response Max. length | 5362 |
| - Response Avg. length | 994 |
| Stage 2 problems | 168K |
| - PLs | 20+ |
| - Images | 168K |
| - Image Height | 26-9694 |
| - Image Width | 125-1484 |
| - Response Max. length | 1978 |
| - Response Avg. length | 400 |

*Figure 3.* PL distribution of M$^2$C-INSTRUCT.

*Table 3.* A comparison of our M$^2$EVAL to other multimodal code datasets.

| Benchmarks | Languages | Eval | Instruct | Sources | #Test | #Train | Task Type |
|---|---|---|---|---|---|---|---|
| Design2Code (Si et al., 2025) | HTML | ✓ | ✗ | Real | 484 | - | Frontend |
| Web2Code (Yun et al., 2024) | HTML | ✓ | ✓ | Synthetic | 5990 | 828K | Frontend |
| MatPlotBench (Yang et al., 2024c) | Python | ✓ | ✗ | Human Curated | 100 | - | Chart-to-Code |
| Plot2Code (Wu et al., 2024) | Python | ✓ | ✗ | Human Curated | 132 | - | Chart-to-Code |
| ChartMimic (Shi et al., 2024) | Python | ✓ | ✗ | Human Curated | 1K | - | Chart-to-Code |
| ChartCoder (Zhao et al., 2025) | Python | ✗ | ✓ | Synthetic | - | 160K | Chart-to-Code |
| SWEbench M (Yang et al., 2024b) | JS | ✓ | ✗ | Human Curated | 619 | - | Issue Resolving |
| Visual SWEbench (Zhang et al., 2024a) | Python | ✓ | ✗ | Human Curated | 128 | - | Chart Issue Resolving |
| MMCode (Li et al., 2024b) | Python | ✓ | ✗ | Crawl | 263 | - | Algorithm |
| HumanEval-V (Zhang et al., 2026) | Python | ✓ | ✗ | Crawl | 253 | - | Algorithm |
| Code-Vision (Wang et al., 2025) | Python | ✓ | ✗ | Human Curated | 438 | - | Algorithm |
| **M$^2$EVAL (Ours)** | 10 PLs | ✓ | ✓ | Human Curated | 300 | 13.1M | Visual Workflow |

aging Qwen2.5-Coder, we generated 12.9 million question-answer pairs, which serve as the foundation for synthesizing multimodal fine-tuning data. In the second stage, following the Magicoder (Wei et al., 2024), we employed two widely used datasets, Evol-CodeAlpaca (Luo et al., 2023) and OSS-Instruct (Wei et al., 2024), to further synthesize multimodal diagram problems.

**Cross Modal Problem.** The Cross-Modal problem refers to converting the code snippets within questions into the visual modality to enhance the code model's capabilities in visual code understanding and Optical Character Recognition (OCR). We utilize the code syntax highlighting tool `Pygments` to render the code within the questions.

**Diagram Problem.** The Diagram-related problem refers to generating corresponding code to solve a problem based on both the textual description and an associated diagram. We propose a two-step synthesis approach to achieve high-quality problem generation. In Step 1, we first prompt Qwen2.5-Coder to generate a diagram based on a standard code problem. We then attempt to render the diagram using Mermaid, filtering out cases where rendering is unsuccessful. In Step 2, we prompt Qwen2.5-Coder to create a multimodal

problem based on the original problem and the diagram generated in Step 1, instructing it to ensure that key information necessary for solving the coding problem is preserved only within the diagram. This guarantees the necessity of the diagram during the problem-solving process.

**Stage-1 Data Synthesis.** Based on the 12.9 million data samples prepared in the first stage, we split the dataset at a 9:1 ratio. Nine parts of the data are used to synthesize Cross-Modal problems, while one part is used to synthesize Diagram-related problems.

**Stage-2 Data Synthesis.** The data from the second stage is entirely used to synthesize Diagram-related problems. A total of 168K problems are synthesized.

### 3.3. M$^2$-CODER Training

Based on our synthesized M$^2$C-INSTRUCT dataset, we propose a two-stage training framework to enhance the model's multimodal code generation capabilities.

**In the first stage**, the model is fine-tuned on extensive M$^2$C-INSTRUCT-stage-1 data. This dataset comprises a large volume of code-related images and textual information,

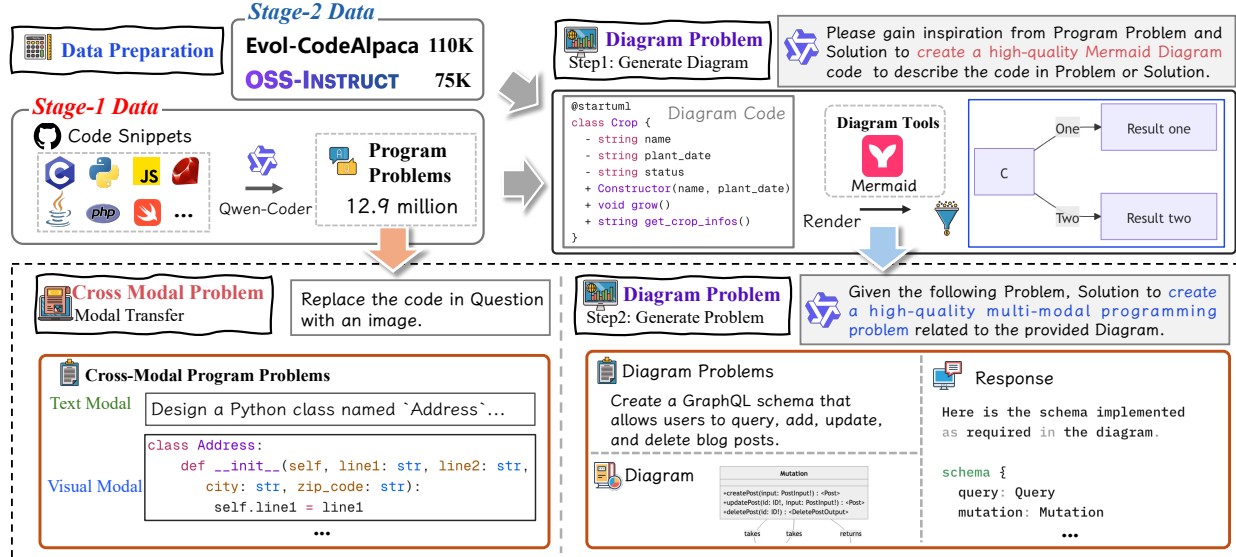

*Figure 4.* The $M^2$C-INSTRUCT data construction pipeline. Stage-1 data generates Cross-Modal Problems by converting code to images. Stage-2 data generates Diagram Problems via a two-step process: (1) creating diagrams from problems/solutions, and (2) formulating multimodal problems that necessitate these diagrams.

a substantial amount of multilingual code, and a smaller subset of diagram-type data. The primary goal of this stage is to establish strong foundational visual understanding and information extraction capabilities, align code with critical multimodal information, and bolster the model's capacity for multilingual code comprehension and generation.

**In the second stage**, building upon the model from the first stage, we further fine-tune it using high-quality $M^2$C-INSTRUCT-stage-2 data. This stage focuses on enhancing the model's image comprehension and instruction-following abilities, particularly for code-related tasks.

## 4. Experiments

### 4.1. Experiment Setup

**Evaluated Models.** In addition to evaluating our $M^2$-CODER, we evaluate 25+ widely used models, including both proprietary and open-source ones. For proprietary models, we assess GPT-4o (OpenAI, 2023), Claude3 (Anthropic, 2024), Gemini-2.5 (Comanici et al., 2025), etc. For open-source models, we test Qwen-VL (Wang et al., 2024a), InternVL (Zhu et al., 2025), Gemma3 (Team et al., 2025), Llama3/4 (Dubey et al., 2024; Meta, 2025), and Phi-3 (Abdin et al., 2024),etc. Moreover, to validate the necessity of including diagrams in $M^2$EVAL, we also evaluate the text-only models Deepseek (Liu et al., 2024a) and Qwen-Coder (Hui et al., 2024), which are particularly strong in code-related tasks.

**$M^2$-CODER Training Setup.** *Stage 1:* We use Qwen2-VL-7B-base to perform full-parameter fine-tuning on the $M^2$C-INSTRUCT-stage-1 dataset (max length 2048 tokens).

Training with 1 epoch using AdamW (LR $5e-5$, batch size 1024), a cosine scheduler (0.1 warmup). *Stage 2:* Initializing from the Stage 1 checkpoint, we conduct fine-tuning only on the LLM (vision tower and projector frozen) using the $M^2$C-INSTRUCT-stage-2 dataset (max length 6000 tokens) for 2 epochs. Other hyperparameters remain identical to Stage 1. (Details can be found in the Appendix C.)

**Evaluation Metrics.** Similar to text-only benchmarks such as HumanEval and MBPP, we adopt the **Pass@k** metric (Chen et al., 2021), which evaluates reliability based on execution results. In this study, we report the greedy **Pass@1** score for all ordinary LLMs using greedy decoding. For thinking type LLM (including Doubao-think, QvQ, and Kimi, etc), we use the corresponding official recommended inference temperature and sampling strategy to achieve the best performance.

### 4.2. Main Results

Table 4 shows the performance of models on $M^2$EVAL for the multilingual multimodal code generation task. The key findings are summarized as follows:

**Multilingual multimodal code generation is challenging.** Even for the strongest model, which also only reached about 50% in the evaluation. This indicates that though most models are trained with text-rich images, the strong text recognition abilities do not guarantee high performance on $M^2$EVAL. Because $M^2$EVAL requires comprehensive OCR, visual logic understanding, and code generation capabilities. In addition, most models perform better in Python, JS, etc., but poorly in C#, Scala, etc. This highlights the great potential for innovation in models, data, or training

*Table 4.* Pass@1 (%) scores of models for multimodal code generation task on M$^2$EVAL. In each model group, the best scores are in bold, and underlined numbers represent the second place. "Avg." represents the average scores of all programming languages.

| Model | Size | Program Languages | | | | | | | | | | Avg. |
|---|---|---|---|---|---|---|---|---|---|---|---|---|
| | | C# | CPP | Java | JS | Kotlin | PHP | Python | Ruby | Scala | Swift | |
| *LLMs w/o Diagram* | | | | | | | | | | | | |
| DeepSeek-V3 | 671B | 0.0 | 0.0 | 0.0 | 0.0 | 0.0 | 0.0 | 0.0 | 0.0 | 0.0 | 0.0 | 0.0 |
| Qwen2.5-Coder | 32B | 0.0 | 0.0 | 0.0 | 0.0 | 0.0 | 0.0 | 0.0 | 0.0 | 0.0 | 0.0 | 0.0 |
| *Proprietary LMMs* | | | | | | | | | | | | |
| GPT-4o | 🔒 | 40.0 | **46.7** | **56.7** | 50.0 | **56.7** | 46.7 | **60.0** | **56.7** | 40.0 | 43.3 | **49.7** |
| Gemini-2.5-Flash-preview | 🔒 | **56.7** | 36.7 | 53.3 | **53.3** | 43.3 | **63.3** | **60.0** | 46.7 | 30.0 | 43.3 | 48.7 |
| Doubao1.5-thinking-pro | 🔒 | 40.0 | 33.3 | 43.3 | 46.7 | 43.3 | 50.0 | **66.7** | 43.3 | 40.0 | 36.7 | 44.3 |
| Claude-3.7-Sonnet | 🔒 | 30.0 | 30.0 | 43.3 | 40.0 | 50.0 | 50.0 | 43.3 | 46.7 | **43.3** | 46.7 | 42.3 |
| Claude-3.5-Sonnet | 🔒 | 46.7 | 26.7 | 40.0 | 40.0 | 26.7 | 53.3 | 53.3 | 46.7 | 40.0 | **50.0** | 42.3 |
| Doubao1.5-vision-pro | 🔒 | 33.3 | **46.7** | 46.7 | 40.0 | 46.7 | 50.0 | 50.0 | 36.7 | 40.0 | 26.7 | 41.7 |
| GPT-4o-mini | 🔒 | 40.0 | 26.7 | 46.7 | 50.0 | 33.3 | 43.3 | 50.0 | 40.0 | 36.7 | 33.3 | 40.0 |
| Qwen-VL-Max | 🔒 | 13.3 | 23.3 | 26.7 | 43.3 | 33.3 | 33.3 | 40.0 | 36.7 | 16.7 | 23.3 | 29.0 |
| *70B+ Open-Weight LMMs* | | | | | | | | | | | | |
| Llama-4-Maverick | 400B | **43.3** | 33.3 | 50.0 | 46.7 | 20.0 | 50.0 | 56.7 | 50.0 | 16.7 | 40.0 | **40.7** |
| Llama-4-Scout | 109B | 26.7 | 26.7 | 50.0 | 30.0 | **23.3** | 36.7 | 53.3 | 30.0 | 16.7 | 16.7 | 31.0 |
| Qwen2-VL-Instruct | 72B | 16.7 | 26.7 | 23.3 | 43.3 | 20.0 | 36.7 | 50.0 | 30.0 | 13.3 | 26.7 | 28.7 |
| QVQ-Preview | 72B | 0.0 | 0.0 | 3.3 | 10.0 | 6.7 | 23.3 | 20.0 | 10.0 | 10.0 | 16.7 | 10.0 |
| Qwen2.5-VL-Instruct | 72B | 16.7 | 26.7 | 23.3 | 26.7 | **23.3** | 33.3 | 30.0 | 30.0 | 16.7 | 20.0 | 24.7 |
| InternVL2-Llama3 | 76B | 13.3 | 13.3 | 23.3 | 23.3 | 20.0 | 26.7 | 23.3 | 23.3 | 10.0 | 16.7 | 19.3 |
| Llama3.2-vision | 90B | 0.0 | 3.3 | 0.0 | 3.3 | 3.3 | 3.3 | 3.3 | 3.3 | 3.3 | 3.3 | 2.7 |
| *16B - 32B Open-Weight LMMs* | | | | | | | | | | | | |
| Gemma-3-it | 27B | **23.3** | **20.0** | **20.0** | 16.7 | **20.0** | **30.0** | **33.3** | 13.3 | 6.7 | **20.0** | **20.3** |
| Qwen2.5-VL-Instruct | 32B | 10.0 | 16.7 | **20.0** | **26.7** | 13.3 | 20.0 | 26.7 | **23.3** | 16.7 | 13.3 | 18.7 |
| Kimi-VL-Thinking | 16B | 0.0 | 13.3 | 13.3 | 20.0 | 6.7 | **30.0** | 26.7 | **23.3** | 6.7 | 3.3 | 14.3 |
| InternVL2 | 26B | 13.3 | 3.3 | 6.7 | 20.0 | 3.3 | 10.0 | 10.0 | 10.0 | 0.0 | 6.7 | 8.3 |
| DeepSeek-VL2 | 27B | 6.7 | 0.0 | 3.3 | 10.0 | 3.3 | 6.7 | 10.0 | 6.7 | 3.3 | 3.3 | 5.3 |
| *4B - 8B Open-Weight LMMs* | | | | | | | | | | | | |
| InternVL3 | 8B | 13.3 | **20.0** | 3.3 | 20.0 | 13.3 | 16.7 | 26.7 | 26.7 | 3.3 | 10.0 | 15.3 |
| Qwen2-VL-Instruct | 7B | 16.7 | 3.3 | 10.0 | 23.3 | 13.3 | 10.0 | 20.0 | 20.0 | 3.3 | 0.0 | 12.0 |
| Qwen2.5-VL-Instruct | 7B | 6.7 | 6.7 | 3.3 | 6.7 | 3.3 | 6.7 | 10.0 | 3.3 | 6.7 | 0.0 | 5.3 |
| Phi3-vision | 4B | 0.0 | 3.3 | 3.3 | 6.7 | 0.0 | 13.3 | 13.3 | 3.3 | 0.0 | 0.0 | 4.3 |
| **M$^2$-CODER (Ours)** | 7B | **26.7** | **20.0** | **20.0** | **30.0** | **16.7** | **36.7** | **36.7** | **33.3** | **16.7** | **16.7** | **25.3** |

objectives to improve the performance of LMMs in multilingual multimodal programming tasks.

**There is still a gap between open and proprietary LMMs.** The results show a clear performance gap between proprietary and open-source models across most programming languages, where models like GPT-4o, Gemini-2.5, Claude-3, and Doubao-1.5 lead the benchmark.

**LLM could not solve problems without diagrams.** We also test the models without diagrams. The results show that DeepSeek-V3 and Qwen2.5-Coder cannot solve the problems without diagrams. This proves that diagrams are necessary in M$^2$EVAL.

**M$^2$-CODER outperforms similarly sized models.** It is noteworthy that M$^2$-CODER-7B outperforms other large language models (LLMs) of similar or even larger sizes (like Qwen2.5-VL-72B). This also validates the effectiveness

of the M$^2$C-INSTRUCT data we have constructed, which can significantly improve the performance of models in multimodal programming tasks.

### 4.3. Further Analysis

We conduct further analyses of the experimental results from multiple perspectives.

**Performance across different tasks** We have classified the problems in M$^2$EVAL into four categories based on their core knowledge points: Class Design, Design Patterns, Algorithms, and Simulation. (Details of this classification can be found in Table 6 in Appendix A.) Our model demonstrably outperforms other models of comparable size across all these task categories. Tasks involving Design Patterns are particularly challenging, as they require models to understand the design patterns in diagrams. Consequently, all

models exhibited relatively poor performance on such tasks.

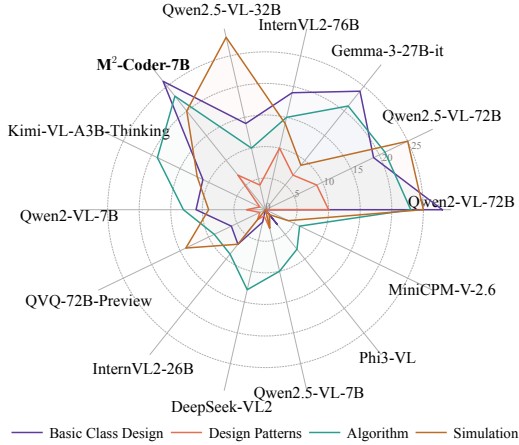

*Figure 5.* Performance comparison of models across task types.

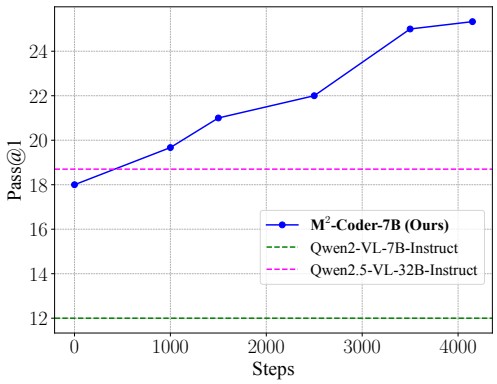

*Figure 6.* Ablation study on stage-1 data efficiency.

**Ablation experiment for M²-CODER fine-tuning.** We perform an ablation study on M²-CODER and present the results in Table 5 and Figure 6. Key findings include:

• **Full Two-Stage SFT is Optimal:** Fine-tuning Qwen2-VL-Base with both stage1 and stage2 SFT (Model ⑦) achieved the highest average Pass@1 score of 25.3.

• **Stage 2 SFT Efficacy:** Applying stage2 SFT alone significantly improved performance. For instance, Qwen2-VL-Instruct (Model ①, Avg. 12.0) + stage2 SFT (Model ②) increased the average to 16.7. Similarly, Qwen2-VL-Base + stage2 SFT (Model ④) reached an average of 18.0.

• **Stage-1 SFT Contribution:** Stage1 SFT on Qwen2-VL-Base (Model ⑥) yielded an average of 10.0. Fine-tuning the vision encoder during stage 1 proved beneficial, as freezing it (Model ⑤) resulted in a lower score of 8.7.

• **Combined Strength:** The results underscore the effectiveness of both SFT stages, with the complete two-stage approach on the base model (Model ⑦) outperforming partial configurations.

• **Stage-1 Data Efficiency:** We select model weights (Model ⑥) preserved at different steps during the stage-1 SFT (1 epoch in total). These checkpoints correspond to varying amounts of stage-1 training data. These selected weights are then used for stage-2 training. It can be observed in Figure 6 that the more data the model is trained on during stage-1, the better its performance after the completion of stage-2 training. This again demonstrates the effectiveness of our stage-1 training.

**Case Study.** Furthermore, we illustrate several error cases in Figure 7. In Case, concerning class implementation, GPT-4o's Java response is missing a function. Qwen-VL does not follow the textual instructions regarding variable visibility requirements, and the code generated by Intern-VL exhibits attribute and function naming inconsistent with the provided diagram. Multimodal code generation tasks demand that models concurrently process both textual and visual information, which requires models to have not only strong visual comprehension skills but also robust code generation capabilities.

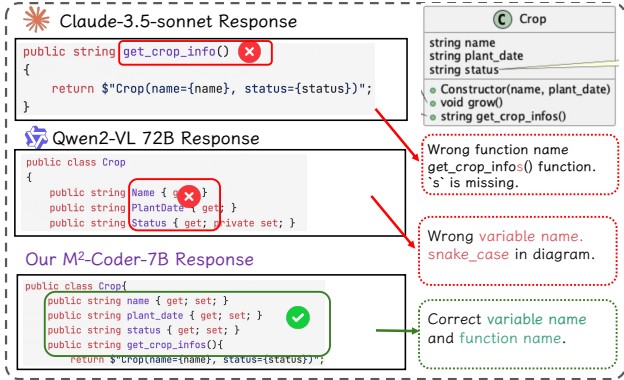

*Figure 7.* A case study for M²EVAL showcasing common error patterns in multimodal code generation.

**Programming Language Matters.** Extending the analysis of language-specific performance in *Error Analysis* section, we present a broader analysis. Figure 8 shows the aggregated correct answers from all evaluated models, grouped by programming language. Scripting languages such as Python, PHP, and JS demonstrated superior performance, contrasting with weaker results from languages like Swift, C#, and Scala. This suggests, firstly, an imbalance in LLM programming proficiency, likely due to benchmark-driven optimizations favoring Python. Secondly, models struggle with languages enforcing strict type checking and rules (e.g., variable and method visibility), which demand more precise instructions and diagram following. Figure 9 details the correct solutions per problem across different languages. On simpler tasks (e.g., problem IDs 1, 2), inter-language performance variance is minimal, with most languages achieving good results. Conversely, for

*Table 5.* Ablation study of our two-stage SFT dataset, M$^2$C-INSTRUCT. Results show Pass@1 on M$^2$EVAL for Qwen2-VL-7B fine-tuned with different parts of M$^2$C-INSTRUCT.

| ID | Model | Program Languages | | | | | | | | | | Avg. |
|---|---|---|---|---|---|---|---|---|---|---|---|---|
| | | C# | CPP | Java | JS | Kotlin | PHP | Python | Ruby | Scala | Swift | |
| ① | Qwen2-VL-Instruct | 16.7 | 3.3 | 10.0 | 23.3 | 13.3 | 10.0 | 20.0 | 20.0 | 3.3 | 0.0 | 12.0 |
| ② | ① + stage2 SFT | 10.0 | 13.3 | 10.0 | 20.0 | 10.0 | 23.3 | 23.3 | 23.3 | **20.0** | 13.3 | 16.7 |
| ③ | Qwen2-VL-Base | - | - | - | - | - | - | - | - | - | - | - |
| ④ | ③ + stage2 SFT | 13.3 | 10.0 | 13.3 | **30.0** | **16.7** | 26.7 | 20.0 | 20.0 | 13.3 | **16.7** | 18.0 |
| ⑤ | ③ + stage1 SFT (freeze) | 10.0 | 6.7 | 13.3 | 16.7 | 3.3 | 3.3 | 16.7 | 13.3 | 3.3 | 0.0 | 8.7 |
| ⑥ | ③ + stage1 SFT | 20.0 | 3.3 | 13.3 | 23.3 | 3.3 | 6.7 | 16.7 | 10.0 | 3.3 | 0.0 | 10.0 |
| ⑦ | ③ + stage1 & stage2 SFT | **26.7** | **20.0** | **20.0** | 30.0 | 16.7 | 36.7 | 36.7 | 33.3 | 16.7 | **16.7** | 25.3 |

medium to difficult problems (e.g., problem IDs 8, 9), these performance disparities are significantly more pronounced.

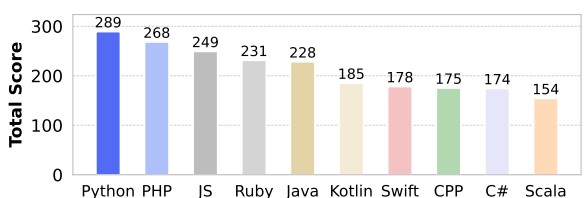

*Figure 8.* Scores on each PLs.

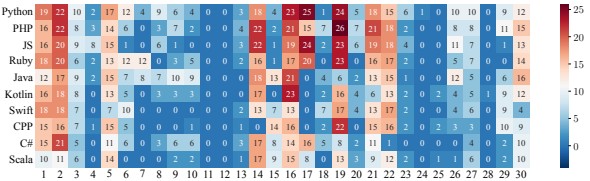

*Figure 9.* Scores on each problem in each languages.

## 5. Related Work

**Code Large Language Models and Evaluation.** The rapid progress of large language models (LLMs)(OpenAI, 2023; Anthropic, 2024; Dubey et al., 2024; Bai et al., 2023a) has enabled significant advances in code-related tasks. Early models like BERT(Devlin et al., 2019) and GPT (Radford et al., 2018) were adapted for code understanding and generation (Chen et al., 2021; Feng et al., 2020; Scao et al., 2022; Yang et al., 2025). More recent Code LLMs benefit from domain-specific pre-training and instruction tuning on large code corpora (Rozière et al., 2023; Luo et al., 2023; Guo et al., 2024; Wei et al., 2024; Lozhkov et al., 2024; Hui et al., 2024), achieving strong performance in tasks like code completion and synthesis. To evaluate these models, diverse benchmarks have emerged, ranging from basic coding tasks (Chen et al., 2021; Austin et al., 2021; Zhuo et al., 2024) to more complex settings such as code repair (Lin et al., 2017; Tian et al., 2024; Zhang et al., 2023; 2025b), multilingual code (Cassano et al., 2023; Athiwaratkun et al., 2023; Chai et al., 2024), repository-level assessments (Liu et al., 2023c; Deng et al., 2024; Bairi et al., 2024), and

agent-based evaluations (Jimenez et al., 2023; Zhang et al., 2024e).

**Visual Reasoning and Code Generation.** Large Multimodal Models(LMMs)(Zhu et al., 2023; Liu et al., 2023a; Bai et al., 2023b; Ye et al., 2023; Liu et al., 2024b; Zhang et al., 2024b; Li et al., 2024a; Wang et al., 2024a; Zong et al., 2024) incorporate visual information into LLMs through visual encoders(Radford et al., 2021), extending the capabilities of LLMs to address visual tasks. With the emergence of powerful visual and semantic capabilities in LMMs and LLMs, many recent works have shifted focus toward more complex multi-modal tasks, such as mathematical reasoning(Trinh et al., 2024; Shao et al., 2024; Huang et al., 2024; Zhang et al., 2024c), chart understanding(Han et al., 2023; Tannert et al., 2023; Singh et al., 2024; Wang et al., 2024b; Li et al., 2024c; Zhang et al., 2024d), code generation(Si et al., 2025; Li et al., 2024b; Wu et al., 2024; Shi et al., 2024), and agent-driven interactions(Zhou et al., 2024; Xie et al., 2024; Liu et al., 2024c; Zhang et al., 2025a). Recent multimodal code works focused on visual algorithmic problems(Li et al., 2024b; Wang et al., 2025), chart code generation(Wu et al., 2024; Shi et al., 2024), and UI design(Si et al., 2025; Liu et al., 2025). Our M$^2$EVAL explores the task of code generation based on code diagrams.

## 6. Conclusion

In this work, we address critical limitations in modern visual-first software development paradigms, particularly emerging vibe coding approaches, by introducing M$^2$C-INSTRUCT, a comprehensive multilingual multimodal instruction-tuning dataset containing over 13.1M samples across 50+ programming languages. We validate our dataset by developing M$^2$-CODER, a multilingual multimodal software developer, and M$^2$EVAL, a novel evaluation benchmark for multimodal code generation. Our experimental results demonstrate that our 7B M$^2$-CODER matches significantly larger 70B+ models, confirming the effectiveness of our training data. Our contributions provide crucial infrastructure for advancing visual-assisted programming AI, representing a significant

step towards more intuitive and effective AI-assisted software development that directly supports vibe coding and visual-interactive development paradigms.

## Acknowledgements

This work is supported by the Fundamental Research Funds for the Central Universitie (Grant No.GW2025-19) and supported by State Key Laboratory of Complex & Critical Software Environment(Grant No. SKLCCSE-2025ZX-26).

## Impact Statement

This paper presents work whose goal is to advance the field of machine learning. There are many potential societal consequences of our work, none of which we feel must be specifically highlighted here.

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

# A. Details of M$^2$EVAL

In this section, we provide a detailed presentation of the M$^2$EVAL statistics, construction process, annotation details, and annotator payment.

## A.1. Detailed Statistics

In Table 6, we list the task categories, the programming knowledge points/concepts tested, the difficulty level, and the corresponding number of test cases for the 30 problems in the dataset (which have parallel versions across 10 programming languages).

*Table 6.* Overview of Test Categories, Difficulties, and Case Counts

| No. | Task | Category | Difficulty | Number of Test Cases |
|---|---|---|---|---|
| 1 | Basic Class Design | Class Design | Easy | 4 |
| 2 | Basic Class Design | Class Design | Easy | 4 |
| 3 | Basic Class Design | Class Design | Medium | 11 |
| 4 | Basic Class Design | Class Design | Hard | 16 |
| 5 | Abstraction and Inheritance | Abstract, Inheritance | Easy | 8 |
| 6 | Abstraction and Inheritance | Interface, Inheritance | Medium | 10 |
| 7 | Creational Patterns | Inheritance, Singleton Pattern | Hard | 8 |
| 8 | Creational Patterns | Abstract, Singleton, Factory Pattern | Hard | 8 |
| 9 | Creational Patterns | Abstract, Singleton, Builder Pattern | Hard | 9 |
| 10 | Structural Pattern | Abstract, Singleton, Adapter Pattern | Hard | 9 |
| 11 | Behavioral Pattern | Abstract, Strategy Pattern, Composite Pattern, Template Method Pattern | Hard | 12 |
| 12 | Creational Patterns | Abstract, Strategy Pattern, Composite Pattern, Template Method Pattern, Abstract Factory Pattern | Hard | 14 |
| 13 | Behavioral Pattern | Abstract, Strategy Pattern, Composite Pattern, Chain of Responsibility Pattern | Hard | 11 |
| 14 | Structural Pattern | Abstract, Proxy Pattern | Easy | 5 |
| 15 | Behavioral Pattern | Abstract, Observer Pattern | Medium | 6 |
| 16 | Algorithm | Size Comparison | Easy | 10 |
| 17 | Algorithm | Binary Search | Easy | 9 |
| 18 | Algorithm | Complex Sorting | Hard | 8 |
| 19 | Algorithm | Sorting | Easy | 10 |
| 20 | Algorithm | Sorting | Hard | 7 |
| 21 | Simulation | Simulation | Easy | 6 |
| 22 | Simulation | Simulation | Easy | 8 |
| 23 | Algorithm | Dynamic Programming, Graph | Hard | 10 |
| 24 | Algorithm | Dynamic Programming | Hard | 12 |
| 25 | Simulation | Simulation, Sorting, Hashing, Collection | Hard | 6 |
| 26 | Simulation | Simulation, Heap, Greedy | Medium | 13 |
| 27 | Simulation | Data Aggregation | Medium | 2 |
| 28 | Simulation | Simulation, Hashing, String Processing | Hard | 9 |
| 29 | Simulation | Rule Simulation | Medium | 6 |
| 30 | Simulation | Sorting, Recommendation | Medium | 19 |

## A.2. M$^2$EVAL Curation Process

**Prototype Problem Design.** To comprehensively evaluate the multimodal coding capabilities of LMMs within a limited set of problems, we prepared the problem design from three dimensions: scenarios, basic programming knowledge, and advanced programming knowledge. First, we designed corresponding scenarios for each problem. Based on these settings, we used an LLM to generate prototype problems in Python, each of which includes a problem prompt, the corresponding solution, and test cases. During this stage, we manually interact with the LLM in multiple rounds to ensure that the prototype

questions align with the problem settings, that the provided solutions are accurate, and that the test cases cover the key evaluation points of each problem.

**Problem Design.** In the second step, based on the identified prototype problems, we further process the Prompt and Solution. Specifically, given the class or function associated with the Solution, we leverage an LLM to generate diagram code in PlantUML or Mermaid. We then manually refine and adjust the generated diagrams to ensure structural and semantic correctness. Subsequently, we revise the Prompt by removing redundant information that overlaps with the diagram content, such that the Prompt alone is insufficient for correct code synthesis. To ensure that the problem remains solvable when both the Prompt and the diagram are provided, we add necessary descriptions directly within the diagram. Through this process, we complete the annotation of Python problems, where each instance consists of a prompt, a diagram, a solution, and corresponding test cases.

**Problem Translate.** Finally, we extend the problems to nine other programming languages. For each problem, the prompt needs to be adapted accordingly (e.g., modifying function names), and both the solution and test cases must be fully translated. We recruited nine volunteers to complete this task. The volunteers conducted the translations with the assistance of LLMs, and during annotation, they were required to verify aspects such as variable consistency and input-output alignment to ensure the accuracy of the translations.

## A.3. Data Annotation

In this section, we detail our data annotation guidelines, the specific division of labor for each step, and the specific remuneration information we pay to the annotators in accordance with relevant regulations.

### A.3.1. DATA ANNOTATION GUIDELINES

Before data annotation, we first designed the following data annotation guidelines to ensure the quality of the data we annotated. The guidelines primarily cover the following aspects:

- **Accessibility:** The reference data we use for annotations comes from materials with permissive licenses. This means it can be freely used and shared for research purposes.

- **Standardized Format:** We provide a sample annotation for 10 programming languages. Annotators are expected to follow this established format for all their annotation work.

- **Difficulty Classification:** Annotators receive clear guidelines on how to categorize the difficulty level for each language. They must strictly apply these guidelines to label problems based on their algorithmic complexity and the features involved.

- **Self-Contained Problems:** Annotators must thoroughly check their annotated problems. This ensures that the problem and visual inputs contain all the necessary information to be solved clearly. The example inputs and outputs must be accurate, the reference solutions should run correctly, and the test cases created must fully assess the correctness of the functions.

### A.3.2. ANNOTATION OF PROTOTYPE PROBLEM AND PYTHON PROBLEM.

The annotation process was conducted collaboratively by two annotators. They jointly investigated and organized the scenarios and knowledge points relevant to each question, and completed the annotation of diagrams, solutions, and test cases with the assistance of LLM. To ensure data quality, mutual cross-checking was performed to verify that each question was answerable and that the intended questions required joint reasoning over both textual and visual inputs. Furthermore, a comprehensive set of test cases was constructed for evaluation purposes. For each question, the annotators also provided a reference solution to further validate the correctness of both the questions and the associated test cases.

### A.3.3. ANNOTATION OF PROBLEM TRANSLATION.

**Employ Annotators.** To translate the annotated Python problems into 9 other programming languages, we employed nine paid annotators for this task. These annotators were Master's or PhD candidates in Computer Science. During the recruitment process, we comprehensively assessed their programming knowledge and skills (e.g., object-oriented programming (OOP) concepts, code compilation, unit testing), as well as their proficiency in using Large Language Models (LLMs).

**Annotation Training.** Each annotator was assigned a programming language with which they were familiar, ensuring they could successfully complete the annotation tasks for their respective language. Furthermore, prior to commencing annotation, all annotators received training. This training detailed the specific annotation requirements and guidelines, and provided them with illustrative examples.

**Data Annotation Content.** In the actual annotation process, with the assistance of LLMs, they were required to translate the functions and parameter descriptions from the original problems into syntactically valid forms for the target language. They also translated the solutions and test cases into the corresponding language. Critically, to ensure the quality of the annotations, the annotators were required to verify that the solutions they translated could successfully pass the test cases they also translated.

A.3.4. QUALITY CONTROL

Subsequent to the completion of their initial annotation tasks, the annotators performed a final data verification phase to ensure the integrity of the data. To facilitate this, each annotator was allocated annotation data produced by two of their peers. Annotators were responsible for examining the correctness of the problems, which involved executing the associated solutions and test cases to validate them, and subsequently correcting any discovered inaccuracies. In instances where errors in the data were contentious, discussions were first initiated with the original annotator. Should an agreement not be reached, a third annotator was introduced to facilitate a joint resolution. The root causes of such errors were then disseminated to the entire annotation team to alert them to analogous potential problems. Ultimately, the canonical solution (labeled by the annotator) was required to pass all corresponding test cases, thereby ensuring the correctness of every problem created (with a 100% pass rate). This meticulous validation process guaranteed the development of a high-quality, multilingual programming benchmark that supports comprehensive analysis and comprehension of code examples across a variety of programming languages.

A.3.5. ANNOTATION PAYMENTS

Annotators were compensated at about $3 per problem, with all work performed online. We also provided the necessary LLM API for the annotation task. In total, approximately 300 problems were annotated. The entire process, including quality assurance for problems (two problems per annotators), incurred a total cost of approximately $2,700.

# B. Details of M²C-INSTRUCT

In this section, we provide a detailed presentation of the M²C-INSTRUCT statistics, construction process, and quality control.

## B.1. Detailed Statistics

**M²C-INSTRUCT Stage 1.**   Figure 7 provides a detailed breakdown of the overall M²C-INSTRUCT stage 1. In addition, specific statistics for the cross-modal and diagram subsets are presented in Figure 8 and Figure 9, respectively.

**M²C-INSTRUCT Stage 1 Lightweight Version.**   Furthermore, we constructed a lightweight subset to facilitate more convenient and low-cost model training. We selected approximately 4.25M data instances with shorter total input-output lengths, the majority of which have a total input-output length of less than 2048 (tokenized with Qwen2-VL (Wang et al., 2024a)). Detailed statistics can be found in Figure 10, Figure 11, and Figure 12.

*Table 7.* M²C-INSTRUCT stage 1 statistics.

| Statistic | Number |
|---|---|
| Total problems | 12962781 |
| Total images | 42360470 |
| Max. Problem length | 4538 |
| Min. Problem length | 3 |
| Avg. Problem length | 452 |
| Max. Response length | 5362 |
| Min. Response length | 5 |
| Avg. Response length | 995 |
| Max. Image height | 13345 |
| Min. Image height | 24 |
| Avg. Image height | 315 |
| Max. Image width | 57246 |
| Min. Image width | 10 |
| Avg. Image width | 619 |

*Table 8.* M²C-INSTRUCT stage 1 cross-modal statistics.

| Statistic | Number |
|---|---|
| Total problems | 11976588 |
| Total images | 41374277 |
| Max. Problem length | 4538 |
| Min. Problem length | 3 |
| Avg. Problem length | 426 |
| Max. Response length | 5362 |
| Min. Response length | 8 |
| Avg. Response length | 1011 |
| Max. Image height | 13345 |
| Min. Image height | 24 |
| Avg. Image height | 291 |
| Max. Image width | 57246 |
| Min. Image width | 10 |
| Avg. Image width | 618 |

*Table 9.* M²C-INSTRUCT stage 1 diagram statistics.

| Statistic | Number |
|---|---|
| Total problems | 986193 |
| Total images | 986193 |
| Max. Problem length | 2041 |
| Min. Problem length | 7 |
| Avg. Problem length | 762 |
| Max. Response length | 1828 |
| Min. Response length | 5 |
| Avg. Response length | 796 |
| Max. Image height | 8018 |
| Min. Image height | 37 |
| Avg. Image height | 1332 |
| Max. Image width | 784 |
| Min. Image width | 153 |
| Avg. Image width | 675 |

*Table 10.* M²C-INSTRUCT lightweight version stage 1 statistics.

| Statistic | Number |
|---|---|
| Total problems | 4249102 |
| Total images | 15582159 |
| Max. Problem length | 1527 |
| Min. Problem length | 3 |
| Avg. Problem length | 420 |
| Max. Response length | 1580 |
| Min. Response length | 50 |
| Avg. Response length | 819 |
| Max. Image height | 8078 |
| Min. Image height | 24 |
| Avg. Image height | 180 |
| Max. Image width | 21859 |
| Min. Image width | 10 |
| Avg. Image width | 509 |

*Table 11.* M²C-INSTRUCT lightweight version stage 1 cross-modal statistics.

| Statistic | Number |
|---|---|
| Total problems | 4112529 |
| Total images | 15445586 |
| Max. Problem length | 1445 |
| Min. Problem length | 3 |
| Avg. Problem length | 413 |
| Max. Response length | 1580 |
| Min. Response length | 50 |
| Avg. Response length | 827 |
| Max. Image height | 8078 |
| Min. Image height | 24 |
| Avg. Image height | 174 |
| Max. Image width | 21859 |
| Min. Image width | 10 |
| Avg. Image width | 509 |

*Table 12.* M²C-INSTRUCT lightweight version stage 1 diagram statistics.

| Statistic | Number |
|---|---|
| Total problems | 136573 |
| Total images | 136573 |
| Max. Problem length | 1527 |
| Min. Problem length | 156 |
| Avg. Problem length | 628 |
| Max. Response length | 1429 |
| Min. Response length | 84 |
| Avg. Response length | 578 |
| Max. Image height | 4084 |
| Min. Image height | 37 |
| Avg. Image height | 806 |
| Max. Image width | 784 |
| Min. Image width | 153 |
| Avg. Image width | 602 |

**M²C-INSTRUCT Stage 2.**

## B.2. M²C-INSTRUCT Construction Process

This section provides a detailed account of the specific methodologies and procedures used for data synthesis and quality control during the construction of M²C-INSTRUCT.

*Table 13.* M²C-INSTRUCT stage 2 statistics.

| Statistic | Number |
|---|---|
| Total problems | 168178 |
| Total images | 168178 |
| Max. Problem length | 2032 |
| Min. Problem length | 31 |
| Avg. Problem length | 277 |
| Max. Response length | 1978 |
| Min. Response length | 8 |
| Avg. Response length | 400 |
| Max. Image height | 9694 |
| Min. Image height | 26 |
| Avg. Image height | 1325 |
| Max. Image width | 1484 |
| Min. Image width | 125 |
| Avg. Image width | 631 |

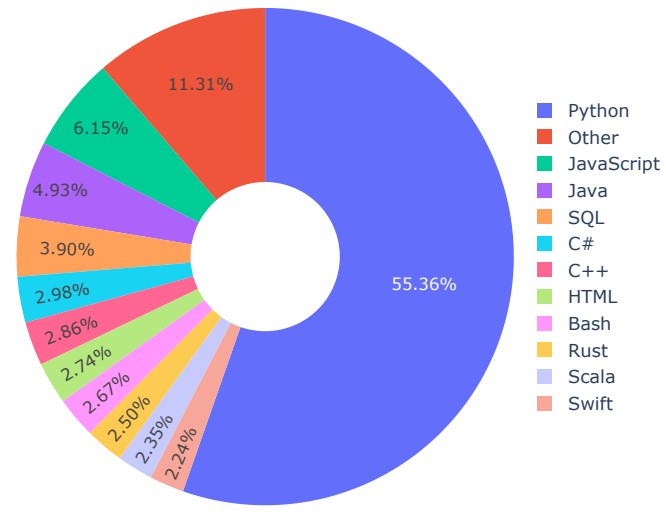

*Figure 10.* Programming language distribution of M²C-INSTRUCT Stage 2.

**Source Data Preparation.**

As illustrated in Figure 4, M²C-INSTRUCT comprises two stages of data preparation for fine-tuning. For Stage 1 of M²C-INSTRUCT, we first collected a large-scale, multilingual code dataset from GitHub, and then sampled and processed the data according to the following steps:

- We preprocessed the data following the StarCoder (Li et al., 2023) data processing pipeline and employed rule-based methods to filter out invalid data (e.g., duplicates, garbled text, illegal information, etc.).

- By prompting Qwen2.5-Coder-32B, we converted the data into a question-answer pair format, generating 12.9 million question-answer pairs. These pairs serve as the foundation for synthesizing multimodal instruction-tuning data.

For the M²C-INSTRUCT stage 2, following the Magicoder (Wei et al., 2024), we employed two widely used datasets, Evol-CodeAlpaca(Luo et al., 2023) and OSS-Instruct(Wei et al., 2024), to further synthesize multimodal diagram problems.

**Cross-Modal Problem.** The Cross-Modal problem refers to converting the code segments within questions into the image modality to enhance the code model's capabilities in visual code understanding and Optical Character Recognition (OCR). We utilize the code syntax highlighting tool `Pygments` to render the code within the questions. Examples of rendered code images are illustrated in Figure 11. The complete cross-modal examples can be found in Appendix D.

**Diagram Problem.** The concept of a 'Diagram-related problem' encompasses tasks that necessitate generating executable code to address a problem, drawing information from both a textual description and an accompanying diagram. We introduce a three-step synthesis methodology designed to produce high-fidelity problems of this nature. In the first step (Step 1), illustrated in Figure 13, Qwen2.5-Coder-32B is initially prompted to generate a diagrammatic representation derived from a standard code problem. Following this, an attempt is made to render the generated diagram using Mermaid, with instances of unsuccessful rendering being systematically excluded. In the second step (Step 2), as shown in Figure 14, Qwen2.5-Coder is again prompted, this time to formulate a multimodal problem. This involves synthesizing the original problem statement with the diagram produced in Step 1. A critical instruction is given to ensure that vital information, indispensable for solving the coding challenge, is exclusively contained within the diagram. This constraint effectively mandates the use of the diagram during the problem-solving phase. In the third step (Step 3), the Mermaid code resulting from Step 2 is employed to render the final visual diagrams with mermaid-cli tools. Illustrative examples of these rendered diagrams are presented in Figure 12. A complete set of examples can be found in Appendix D.

```bash
1   #!/bin/bash
2
3   # Check if the correct number of arguments is provided
4   if [ "$#" -ne 2 ]; then
5       echo "Usage: $0 <file_path> <description>"
6       exit 1
7   fi
8
9   # Assign arguments to variables
10  FILE_PATH=$1
11  DESCRIPTION=$2
12
13  # Check if the file exists
14  if [ ! -f "$FILE_PATH" ]; then
15      echo "Error: File does not exist."
16      exit 1
17  fi
```

```javascript
function applyProxySettings(userGroups, groupPolicies) {
  let effectiveSettings = {};

  // Iterate over each group the user belongs to
  for (let group of userGroups) {
    // Check if the group has any proxy settings defined
    if (groupPolicies[group] && groupPolicies[group].proxy) {
      // Update the effective settings with the current group'
      effectiveSettings = { ...groupPolicies[group] };
    }
  }

  return effectiveSettings;
}

// Example usage
const userGroups = ['Admins', 'Developers', 'Managers'];
const groupPolicies = {
  Admins: { proxy: 'http://admin-proxy:8080' },
  Developers: { proxy: 'http://dev-proxy:8080' },
  Managers: { proxy: 'http://manager-proxy:8080' }
};
```

```java
import java.util.Scanner;

public class JavaLOCCounter {
    public static void main(String[] args) {
        Scanner scanner = new Scanner(System.in);
        int totalJavaLines = 0;

        // Read input line by line
        while (scanner.hasNextLine()) {
            String line = scanner.nextLine();
            String[] parts = line.split(" ");

            // Check if the file is a Java file
            if (parts[0].endsWith(".java")) {
                // Sum up blank, comment, and code lines
                int blankLines = Integer.parseInt(parts[1]);
                int commentLines = Integer.parseInt(parts[2]);
                int codeLines = Integer.parseInt(parts[3]);

                totalJavaLines += blankLines + commentLines + codeLines;
            }
        }
    }
}
```

```javascript
const myCamera = new Camera();
const photo1 = myCamera.takePhoto();
photo1.applyFilter('grayscale');
photo1.save(); // Should log: Photo ID: 1,

const photo2 = myCamera.takePhoto();
photo2.applyFilter('sepia');
photo2.save(); // Should log: Photo ID: 2,
```

```javascript
class EncryptionModule {
    private modules: { [id: number]: string } = {};
    private nextID: number = 1;

    // Method to open a new encryption module
    openModule(algorithm: string): number {
        const id = this.nextID++;
        this.modules[id] = algorithm;
        console.log(`Module ${id} opened with algorithm ${algorithm}`);
        return id;
    }
}
```

*Figure 11.* Examples of rendered code images via pygments.

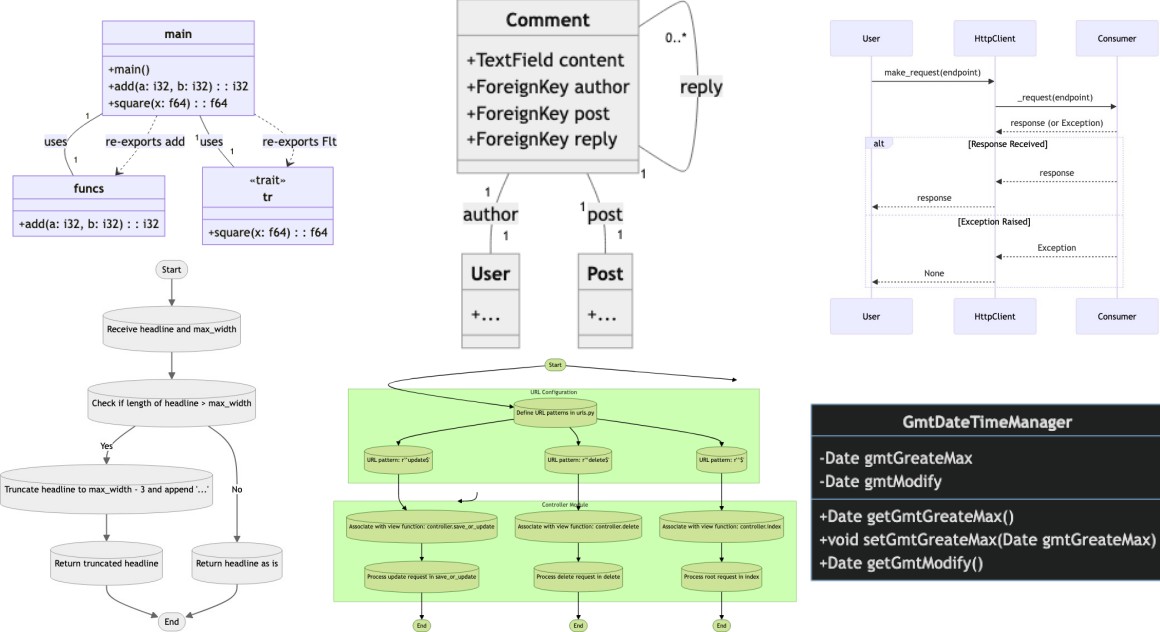

*Figure 12.* Examples of diagram code images via mermaid.

---

## Prompt Template For Step 1

### Please gain inspiration from Program Problem and Solution to create a high-quality **Mermaid Diagram code** to describe the code in Problem or Solution.

#### Problem for inspiration:
{problem}

#### Solution:
{solution}

#### Guidelines for creating mermaid diagram code:
1. Identify and highlight the key information from the problem and solution to include in the Mermaid diagram, such as main processes, detail args, steps, decision points, and outcomes.
2. Use Mermaid syntax to visually represent the flow of the problem and solution. For example, you can use **flowcharts, sequence diagrams, or class diagrams .etc** depending on the complexity and nature of the problem and solution.
3. Place the Mermaid code inside ```mermaid```.
4. Description in the diagram should be natural, here is an example:

```
flowchart TD
    Start([Start]) --> Input[("Receive arrays A and B of length n")]
    Input --> Init[("Initialize result array C with size 2n-1")]
    Init --> Loop["Iterate i from 0 to 2n-2"]
    Loop --> InnerLoop["For each i, iterate j from max(0, i-n+1) to min(i, n-1)"]
    InnerLoop --> Compute[("Add A[j] * B[i-j] to C[i]")]
    Compute --> InnerLoop
    InnerLoop -- End --> Loop
    Loop -- End --> Output[("Output array C")]
    Output --> End([End])
```

5. Most Importantly, **ensuring all text is described within quotes** to avoid syntax errors.

*Figure 13.* Prompt template for step 1 of constructing diagram-type data in M$^2$C-INSTRUCT.

---

## Prompt Template For Step 2

### Please gain inspiration from the following Problem, Solution to create a high-quality **multi-modal** programming problem related to the **provided Diagram**. Present your output in two distinct sections: [Incomplete Problem] and [Solution].

#### Problem for inspiration:
{problem}

#### Solution for inspiration:
{solution}

#### Diagram:
{mermaid_code}

#### Guidelines for each section:
**1. [Incomplete Problem]:**
- This problem is incomplete, and can not be solved with only your generated [Incomplete Problem], some key information is only provided by the diagram.
- Replace some steps or details with "xxx is detailed in the provided diagram" or "xxx could be found in the diagram".
- Ensure [Incomplete Problem] is incomplete, must refer to the diagram for supplementary information.
- Don't need to include the diagram code in the problem, the diagram will be provided as an image.

**2. [Solution]:**
- Offer a comprehensive, **correct** solution that accurately addresses the [Problem] you provided.
- Don't generate the main or check function.

*Figure 14.* Prompt template for step 2 of constructing diagram-type data in M$^2$C-INSTRUCT.

## C. Experimental Details

### C.1. $M^2$-CODER Training

In this section, we list in detail the hyperparameters we set during the $M^2$-CODER training. Our training tasks utilized 8 NVIDIA A800 GPUs to complete all experiments. All training works are done with the open-source training framework LlamaFactory(Zheng et al., 2024b).

**Stage 1 fine-tuning** We initialize our model from the Qwen2-VL-7B-base. Stage 1 fine-tuning is performed using a full fine-tuning approach, where the vision tower, multi-modal projector, and language model are all unfrozen and trained. Training is conducted on the $M^2$C-INSTRUCT-stage-1. Input data is formatted using the qwen2_vl template, with sequences truncated or padded to a maximum length of 2048 tokens. For optimization, we employ AdamW (Loshchilov & Hutter, 2017) as the optimizer, with a batch size of 1024 and a maximum sequence length of 2048. The model is trained for 1.0 epoch with a learning rate of $5 \times 10^{-5}$. A cosine learning rate scheduler is employed with a warmup ratio of 0.1. Training is performed with bfloat16 (bf16) mixed precision and accelerated using DeepSpeed with a ZeRO Stage 2 configuration.

**Stage 2 fine-tuning** In Stage 2, we initialize our model from the Stage 1 checkpoint and continue with SFT. During this stage, the vision tower and multi-modal projector are kept frozen, and only the language model parameters are fine-tuned. Training is conducted on the $M^2$C-INSTRUCT-stage-2. Input data is formatted using the qwen2_vl template, with sequences truncated or padded to a maximum length of 6000 tokens. For optimization, we employ AdamW as the optimizer, with a global batch size of 1024 and a maximum sequence length of 6000. The model is trained for 2.0 epochs with a learning rate of 5e-5. A cosine learning rate scheduler with a warmup ratio of 0.1 is utilized. Training is performed with bfloat16 mixed precision and accelerated using DeepSpeed with a ZeRO Stage 2 configuration.

### C.2. Evaluation Setup

**Evaluation Environment.** Table 14 presents the code sandbox environment used in our evaluation, including version information for 10 programming languages. The construction of the environment refers to MultiPL-E(Cassano et al., 2023). We will release this environment as an open-source Dockerfile to facilitate future evaluation and testing.

*Table 14.* Runtime environments for 10 programming languages.

| Language | Runtime Environments |
| --- | --- |
| C# | Mono C# compiler version 6.12.0.200 |
| CPP | g++ (Ubuntu 11.4.0-1ubuntu1 22.04) 11.4.0 |
| Java | openjdk version "11.0.26" 2025-01-21 |
| JavaScript | Node.js v23.10.0 |
| Kotlin | kotlinc-jvm 2.1.10 (JRE 11.0.26+4-post-Ubuntu-1ubuntu122.04) |
| PHP | PHP 8.1.2-1ubuntu2.20 (cli) (built: Dec 3 2024 20:14:35) (NTS) |
| Python | Python 3.10.12 |
| Ruby | ruby 3.0.2p107 (2021-07-07 revision) [x86_64-linux-gnu] |
| Scala | Scala code runner version 3.3.5 |
| Swift | Swift version 6.0.3 (swift-6.0.3-RELEASE) |

### C.3. Additional Analysis

Figure 15 reveals that our $M^2$-CODER model demonstrates competitive performance, with its 7B parameters achieving results comparable to some models around the 70B scale, particularly in Basic Class Design tasks. Concurrently, leading models such as GPT-4o and Gemini-2.5-flash exhibit a significant lead in Design Patterns tasks. This disparity underscores the critical importance of underlying programming knowledge and reasoning for tasks involving diagrams, as proficiency in areas like Design Patterns, which often rely on diagrammatic representation, is a key differentiator and highlights a crucial aspect for understanding and generating code from visual specifications.

Figure 16 shows the model's performance on different problems (a score of 10 indicates that the code in all languages passes the tests). It can be seen that on some simple problems (e.g., problems 1, 2, 16, and 19), most models perform well in all languages. However, on some complex problems, the model's performance is relatively poor. This indicates that even for the same problem, the choice of programming language for the solution significantly impacts accuracy.

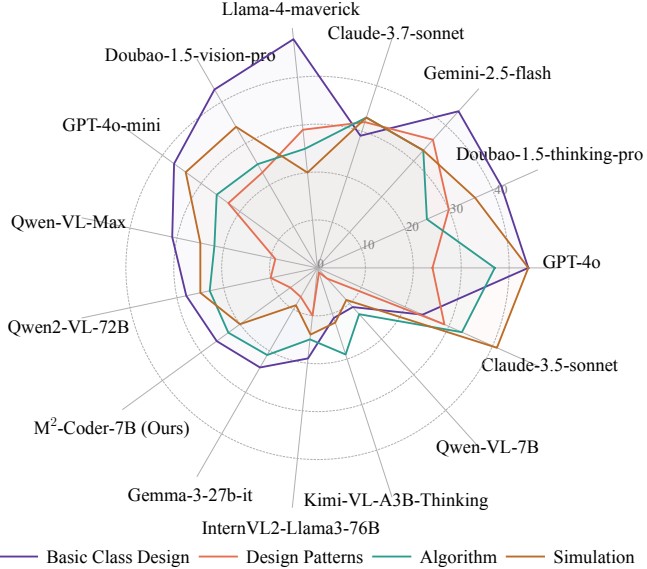

*Figure 15.* Performance comparison of models across task types.

## C.4. Cases for Error Analysis

In Figure 17 and Figure 18, we present illustrative error cases pertaining to visual understanding and programming language generation by GPT-4o, one of today's leading Large Multimodal Models (LMMs). Despite its advanced capabilities, even this model makes some fundamental mistakes. For instance, Case 1 involves errors in variable and function naming; Case 2 exhibits instruction non-compliance; and Cases 3 and 4 reveal intrinsic language-specific errors. These examples underscore the multifaceted complexity of multimodal programming, highlighting persistent and significant challenges in areas such as the precise capture of visual information, robust instruction following, and fundamental programming language proficiency.

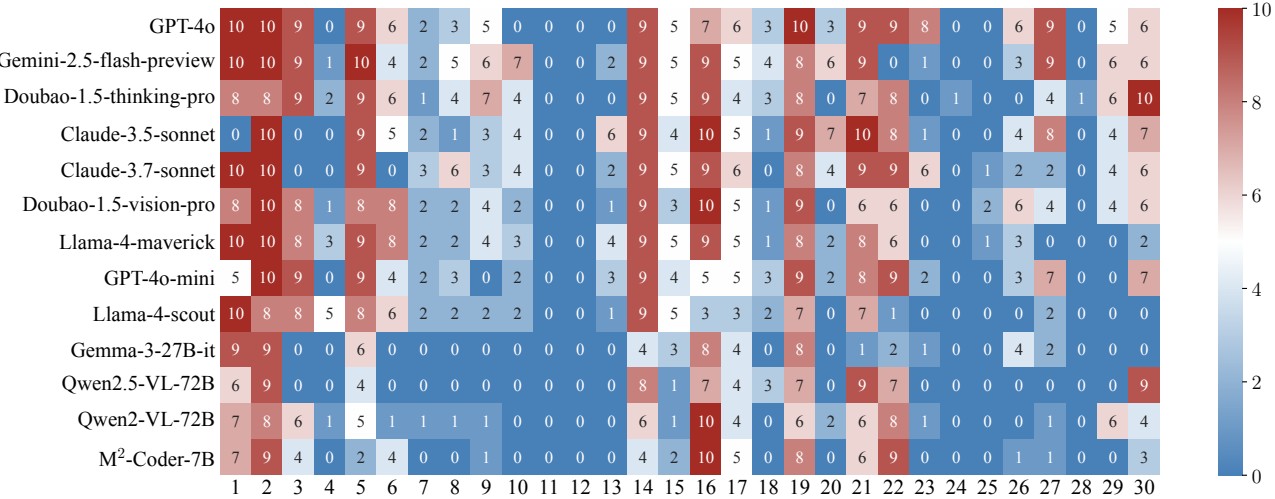

*Figure 16.* Scores of models on each problem.

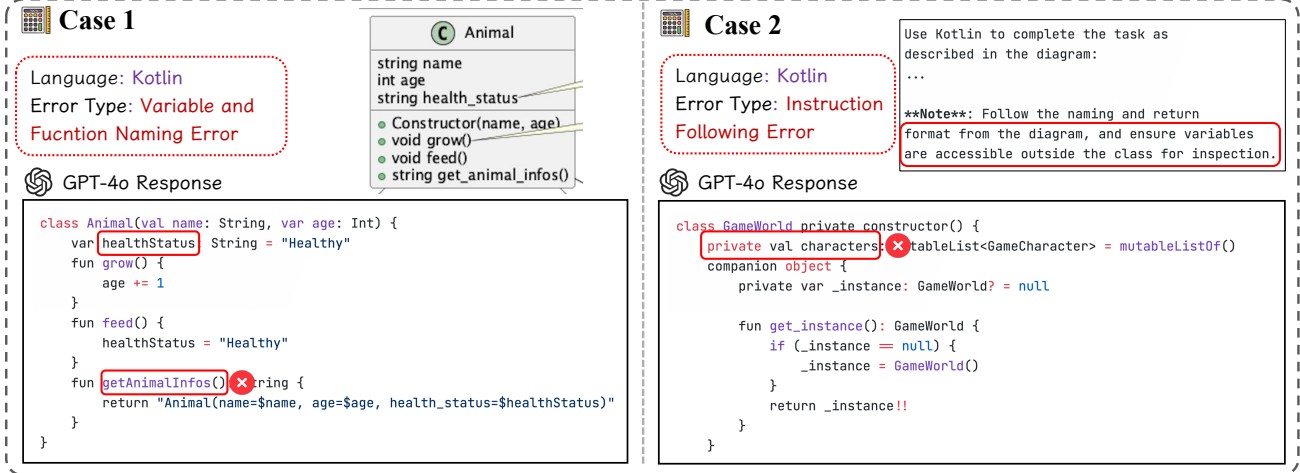

*Figure 17.* Kotlin error cases from GPT-4o on M²EVAL.

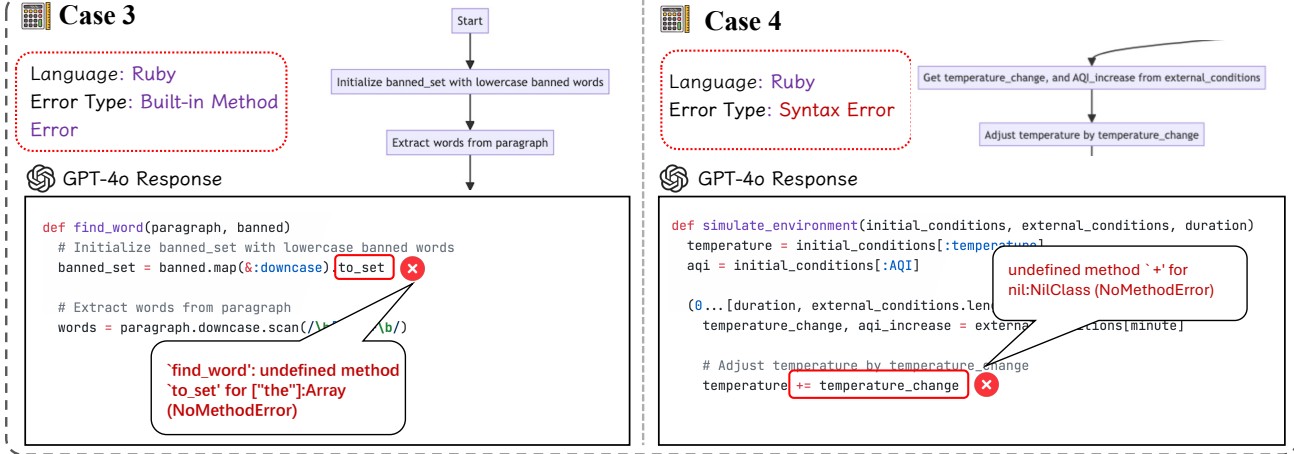

*Figure 18.* Ruby error cases from GPT-4o on M²EVAL.

# D. Examples

## D.1. M$^2$C-INSTRUCT Examples

---

### M$^2$C-INSTRUCT Cross-Modal Example 1

#### Problem

You are tasked with writing a TypeScript program that simulates a simple file system on a hard drive. The file system should support basic operations such as creating files, deleting files, and listing files. Each file will have a name and a size. The hard drive has a fixed capacity, and the program should handle cases where the hard drive is full.

### Requirements
1. **Create File**: Add a file to the file system. If the hard drive is full, return an error message.
2. **Delete File**: Remove a file from the file system.
3. **List Files**: Return a list of all files in the file system along with their sizes.
4. **Check Capacity**: Return the remaining capacity of the hard drive.

### Example

```typescript
const fs = new FileSystem(100); // 100 MB hard drive

fs.createFile("file1.txt", 20); // Success
fs.createFile("file2.txt", 30); // Success
fs.createFile("file3.txt", 60); // Error: Not enough space

fs.listFiles(); // ["file1.txt (20 MB)", "file2.txt (30 MB)"]
fs.deleteFile("file1.txt");
fs.listFiles(); // ["file2.txt (30 MB)"]
fs.checkCapacity(); // 70 MB
```

#### Response

Here is an improved version of your code with these considerations:

```typescript
class File {
    constructor(public name: string, public size: number) {
        if (size <= 0) {
            throw new Error("File size must be a positive number");
        }
    }
}
class FileSystem {
    private files: File[] = [];
    private totalCapacity: number;
    private usedCapacity: number = 0;

    constructor(capacity: number) {
...
```

*Figure 19.* Cross-modal example 1 from M$^2$C-INSTRUCT.

## M$^2$C-INSTRUCT Cross-Modal Example 2

### Problem

You are tasked with simulating the behavior of an Infrared LED (IR LED) in a security system. The IR LED has a specific power consumption and brightness level, which changes based on the ambient temperature. The goal is to create a JavaScript function that calculates the brightness of the IR LED given the ambient temperature and the power consumption. The brightness is calculated using the formula:

$$\text{Brightness} = \text{Power Consumption} \times (1 - \frac{\text{Ambient Temperature} - 25}{100})$$

Where: - Power Consumption is in milliwatts (mW). - Ambient Temperature is in degrees Celsius (°C). - Brightness is in arbitrary units (AU). ...

```javascript
1   /**
2    * Function to calculate the brightness of an IR LED based on power consumption and ambient temperature.
3    * @param {number} powerConsumption - Power consumption of the IR LED in milliwatts (mW).
4    * @param {number} ambientTemperature - Ambient temperature in degrees Celsius (° C).
5    * @returns {number} - Brightness of the IR LED in arbitrary units (AU).
6    */
7   function calculateBrightness(powerConsumption, ambientTemperature) {
8       // Calculate the brightness using the given formula
9       let brightness = powerConsumption * (1 - (ambientTemperature - 25) / 100);
10
11      // Ensure the brightness does not go below zero
12      if (brightness < 0) {
13          brightness = 0;
14      }
15
16      return brightness;
17  }
18
19  // Example usage:
20  console.log(calculateBrightness(100, 25)); // Output: 100
21  console.log(calculateBrightness(100, 75)); // Output: 50
22  console.log(calculateBrightness(100, -25)); // Output: 125
23  console.log(calculateBrightness(100, 125)); // Output: 0
```

### Response

The task is to simulate the behavior of an Infrared LED (IR LED) in a security system by calculating its brightness based on the power consumption and ambient temperature. The brightness is calculated using a specific formula, and we need to ensure that the brightness does not go below zero.
Here's the JavaScript function 'calculateBrightness' that performs this calculation:

```javascript
/**
 * Function to calculate the brightness of an IR LED
 based on power consumption and ambient temperature.
 * @param {number} powerConsumption - Power consumption of the IR LED in milliwatts (mW).
 * @param {number} ambientTemperature - Ambient temperature in degrees Celsius (°C).
 * @returns {number} - Brightness of the IR LED in arbitrary units (AU).
 */
function calculateBrightness(powerConsumption, ambientTemperature) {
    // Calculate the brightness using the given formula
    let brightness = powerConsumption * (1 - (ambientTemperature - 25) / 100);

    // Ensure the brightness does not go below zero
    if (brightness < 0) {
        brightness = 0;
    }

    return brightness;
}

// Example usage:
console.log(calculateBrightness(100, 25));  // Output: 100
console.log(calculateBrightness(100, 75));  // Output: 50
console.log(calculateBrightness(100, -25)); // Output: 125
console.log(calculateBrightness(100, 125)); // Output: 0
...
```

*Figure 20.* Cross-modal example 2 from M$^2$C-INSTRUCT.

## M²C-INSTRUCT Cross-Modal Example 3

### Problem

You are tasked with creating a JavaScript factory function that generates objects representing database records. This factory function should allow for the creation of objects with specific properties and methods, similar to how a plugin might extend a class in Perl's Class::DBI::Plugin::Factory. The factory should be able to handle different types of database records, such as 'User' and 'Product', each with their own set of properties and methods.
### Requirements
1. **Factory Function**: Create a factory function 'createRecordFactory' that takes a 'type' parameter and returns a constructor function for that type of record.
2. **Constructor Function**: The constructor function should initialize the object with properties based on the type.
3. **Methods**: Each type of record should have specific methods. For example, 'User' records should have a 'greet' method, and 'Product' records should have a 'displayPrice' method.
4. **Edge Cases**: Handle cases where the type is not recognized by returning a generic 'Record' type with a default method.
### Example Usage

```javascript
1    // Create a User factory
2    const UserFactory = createRecordFactory('User');
3    const user = new UserFactory('Alice', 'alice@example.com');
4    console.log(user.greet()); // Output: Hello, Alice!
5
6    // Create a Product factory
7    const ProductFactory = createRecordFactory('Product');
8    const product = new ProductFactory('Laptop', 999.99);
9    console.log(product.displayPrice()); // Output: Price: $999.99
10
11   // Create a generic Record factory for unrecognized types
12   const UnknownFactory = createRecordFactory('Unknown');
13   const unknown = new UnknownFactory();
14   console.log(unknown.info()); // Output: This is a generic record.
```
...

### Response

```
### Explanation and Code Review

The provided solution effectively meets the requirements by creating a factory
function that generates constructor functions for different types of database records.
Here's a detailed breakdown of the solution and some suggestions for improvement:

#### Key Components

1. **Factory Function (`createRecordFactory`)**:
   - This function takes a `type` parameter and returns a constructor function for the specified type.
   - It uses a switch statement to determine which constructor to return.

2. **Constructor Functions**:
   - **User**: Initializes with `name` and `email` properties and has a `greet` method.
   - **Product**: Initializes with `name` and `price` properties and has a `displayPrice` method.
   - **Record**: A generic constructor for unrecognized types with an `info` method.

3. **Prototypes**:
   - Methods are added to the prototypes of the constructors to ensure they are shared across
   all instances, which is a good practice for memory efficiency.

4. **Edge Cases**:
   - If the `type` is not recognized, the factory returns a `Record` constructor, providing
   a default method.
```

*Figure 21.* Cross-modal example 3 from M²C-INSTRUCT.

## M²C-INSTRUCT Diagram Example 1

### Problem

You are tasked with creating a TypeScript class method that fetches and processes data from a server using the provided code snippet as a reference. Your task is to implement the 'jsonMutationDataFetcher' function and ensure that the 'fetchData' method correctly handles the asynchronous data retrieval and processing.

The 'fetchData' method is a private method of a TypeScript class and is decorated with '@autobind'. It takes a 'filter' parameter of type 'TFilter' and returns a 'Promise' of type 'TData'. The 'jsonMutationDataFetcher' function is used to fetch and process the data. It takes three parameters: 'mutationName' of type 'string', 'filter' of type 'TFilter' or its formatted version, and 'formatter' function.

Your task is to implement the 'jsonMutationDataFetcher' function and ensure that the 'fetchData' method correctly calls it with the appropriate parameters. Additionally, you need to handle the asynchronous nature of data fetching and processing. ...

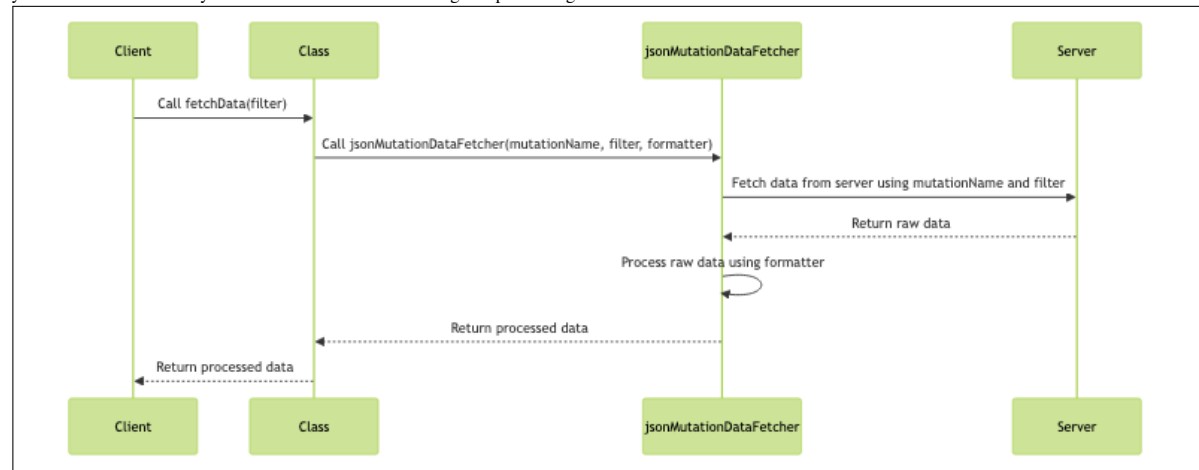

### Response

```
// Solution for jsonMutationDataFetcher function
async function jsonMutationDataFetcher<TRaw, TData>(
  mutationName: string,
  filter: TFilter,
  formatter: (data: TRaw) => TData
): Promise<TData> {
  // Fetch data from the server using the mutationName and filter
  const rawData = await fetchDataFromServer(mutationName, filter);
  // Process the raw data using the provided formatter function
  const processedData = formatter(rawData);
  return processedData;
}

// Solution for fetchData method
@autobind
private async fetchData(filter: TFilter): Promise<TData> {
  return jsonMutationDataFetcher<TRaw, TData>(
    this.props.mutationName,
    this.props.filterFormatter ? this.props.filterFormatter(filter) : filter,
    this.props.formatter
  );
}
...
```

In the solution, the 'jsonMutationDataFetcher' function is implemented to handle the data fetching and processing. It takes the 'mutationName', 'filter', and 'formatter' as parameters and returns a 'Promise' of type 'TData'. ...

*Figure 22.* Diagram example 1 from M²C-INSTRUCT.

## M²C-INSTRUCT Diagram Example 2

### Problem

You are given a list of software applications installed on a user's computer. The list contains the names of the applications, but there are some errors in the names. The errors include misspellings, incorrect capitalization, and missing characters. Your task is to create a Python function that takes the list of applications as input and returns a corrected list with the errors fixed.

The function signature is: "'python def fix_application_names(apps: list) -¿ list: pass "' ...

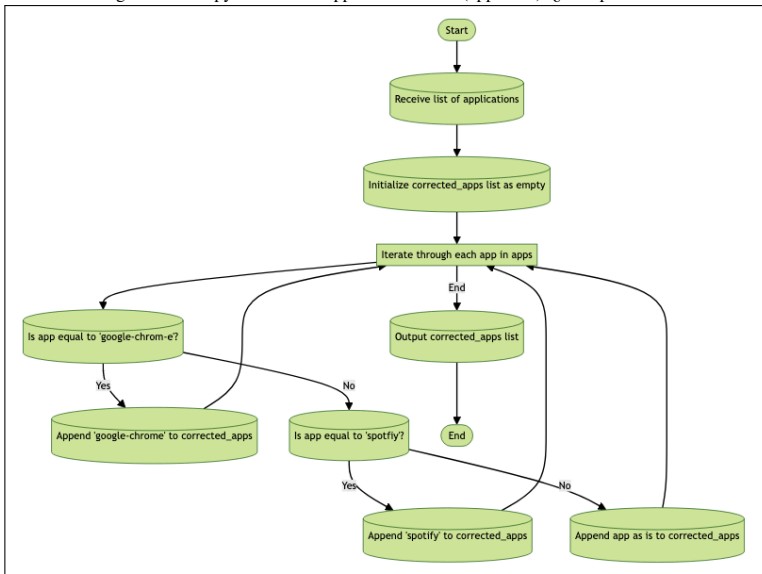

### Response

```python
def fix_application_names(apps: list) -> list:
    corrected_apps = []
    for app in apps:
        if app == "google-chrom-e":
            corrected_apps.append("google-chrome")
        elif app == "spotfiy":
            corrected_apps.append("spotify")
        else:
            corrected_apps.append(app)
    return corrected_apps
```

The 'fix_application_names' function iterates through the input list of applications and corrects the errors by replacing the incorrect names with the correct ones. Specifically, it handles the corrections for "google-chrom-e" and "spotfiy" and appends the corrected names to a new list, which is then returned as the output. The detailed flow of this process can be found in the provided diagram.

*Figure 23.* Diagram example 2 from M²C-INSTRUCT.

## M²C-INSTRUCT Diagram Example 3

### Problem

Your task is to implement the 'DebugContextHolder' interface and provide a concrete implementation of the 'getDebugContext' method. Additionally, you need to create a sample usage scenario to demonstrate the functionality of the debugging context holder.

Your implementation should adhere to the following requirements: - Implement the 'DebugContextHolder' interface with the 'getDebugContext' method. - Create a concrete implementation of the 'DebugContextHolder' interface. - Demonstrate the usage of the debugging context holder in a multi-threaded environment.

You can use the given code snippet as a reference for the interface definition and method signature. The class structure and relationships are detailed in the provided diagram. ...

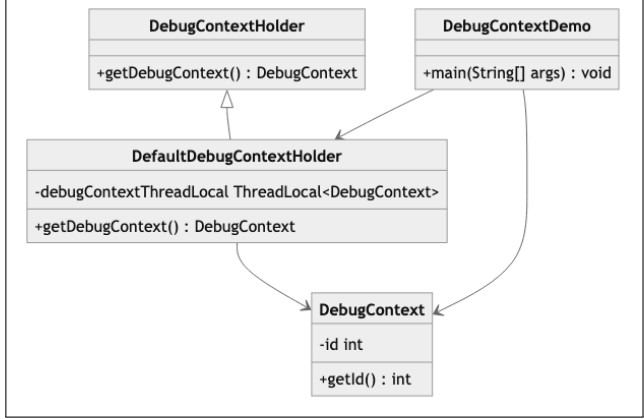

### Response

```java
import java.util.concurrent.atomic.AtomicInteger;

/**
 * Debug context holder interface. By default debugging context stores in ThreadLocal
 variable {@link DefaultDebugContextHolder}

public interface DebugContextHolder {

    /**
     * Get debug context.
     *
     * @return DebugContext
     */
    DebugContext getDebugContext();
}

/**
 * Concrete implementation of DebugContextHolder using ThreadLocal for storing debugging context.
 */
public class DefaultDebugContextHolder implements DebugContextHolder {

    private static final ThreadLocal<DebugContext> debugContextThreadLocal =
    new ThreadLocal<DebugContext>() {
        @Override
        protected DebugContext initialValue() {
            return new DebugContext();
        }
    };
...
```

In this solution, we first define the 'DebugContextHolder' interface with the 'getDebugContext' method. Then, we provide a concrete implementation, 'DefaultDebugContextHolder', which uses a 'ThreadLocal' variable to store the debugging context. We also demonstrate the usage of the debugging context holder in a multi-threaded environment by creating and starting multiple threads to access the debugging context. Finally, we define the 'DebugContext' class to represent the debugging context with a unique ID for each instance. The class structure and relationships are as detailed in the provided diagram.

*Figure 24.* Diagram example 3 from M²C-INSTRUCT.

## M²C-INSTRUCT Diagram Example 4

### Problem

You are tasked with implementing a multi-modal C++ class that encapsulates a set of standard functions and global variables for LLVM (Low Level Virtual Machine) operations. The provided code snippet is a part of a namespace 'asllvm::detail' containing two structs: 'StandardFunctions' and 'GlobalVariables'. The 'StandardFunctions' struct holds references to various LLVM function callees, while the 'GlobalVariables' struct is incomplete in the given snippet.

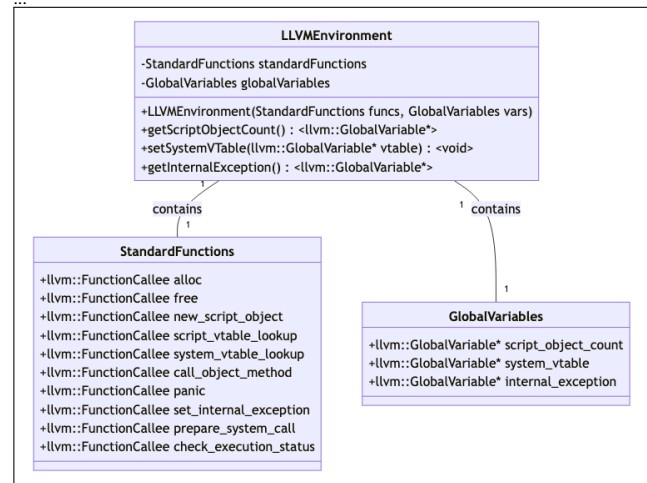

### Response

```cpp
#include <llvm/IR/Function.h>
#include <llvm/IR/GlobalVariable.h>

namespace asllvm::detail{
    struct StandardFunctions{
        llvm::FunctionCallee alloc;
        llvm::FunctionCallee free;
        llvm::FunctionCallee new_script_object;
        llvm::FunctionCallee script_vtable_lookup;
        ...
    };
    struct GlobalVariables{
        llvm::GlobalVariable* script_object_count;
        ...
    };

    class LLVMEnvironment{
    private:
        StandardFunctions standardFunctions;
        GlobalVariables globalVariables;

    public:
        // Constructor to initialize standard functions and global variables
        LLVMEnvironment(StandardFunctions funcs, GlobalVariables vars)
            : standardFunctions(funcs), globalVariables(vars) {}

        // Method to get the script object count
        llvm::GlobalVariable* getScriptObjectCount() const{
            return globalVariables.script_object_count;
        }
        // Method to set the system vtable
        void setSystemVTable(llvm::GlobalVariable* vtable){
            globalVariables.system_vtable = vtable;
        }
    }

...
```

In this solution, the 'GlobalVariables' struct is completed by adding member variables for 'script_object_count', 'system_vtable', and 'internal_exception'. Additionally, a C++ class 'LLVMEnvironment' is defined to encapsulate the 'StandardFunctions' and 'GlobalVariables' structs. The class provides methods to access and manipulate the standard functions and global variables, demonstrating encapsulation and object-oriented programming principles. The relationships and specific details of the classes and their methods are consistent with the provided diagram.

*Figure 25.* Diagram example 4 from M²C-INSTRUCT.

## M²C-INSTRUCT Diagram Example 5

### Problem

You are tasked with creating a multi-modal Python class that can serialize and deserialize data for a pet adoption application. The application needs to handle various attributes of pets, such as their name, age, breed, and availability for adoption. Your task is to implement a serializer class that can convert pet objects into JSON format for storage and transmission, as well as deserialize JSON data back into pet objects.

You are provided with a basic serializer class, 'PetSerializer', which needs to be extended and customized to handle the specific attributes of pets. The 'PetSerializer' class should include methods for serializing and deserializing pet objects, ensuring that the data is accurately represented in JSON format and can be reconstructed into pet objects.

...

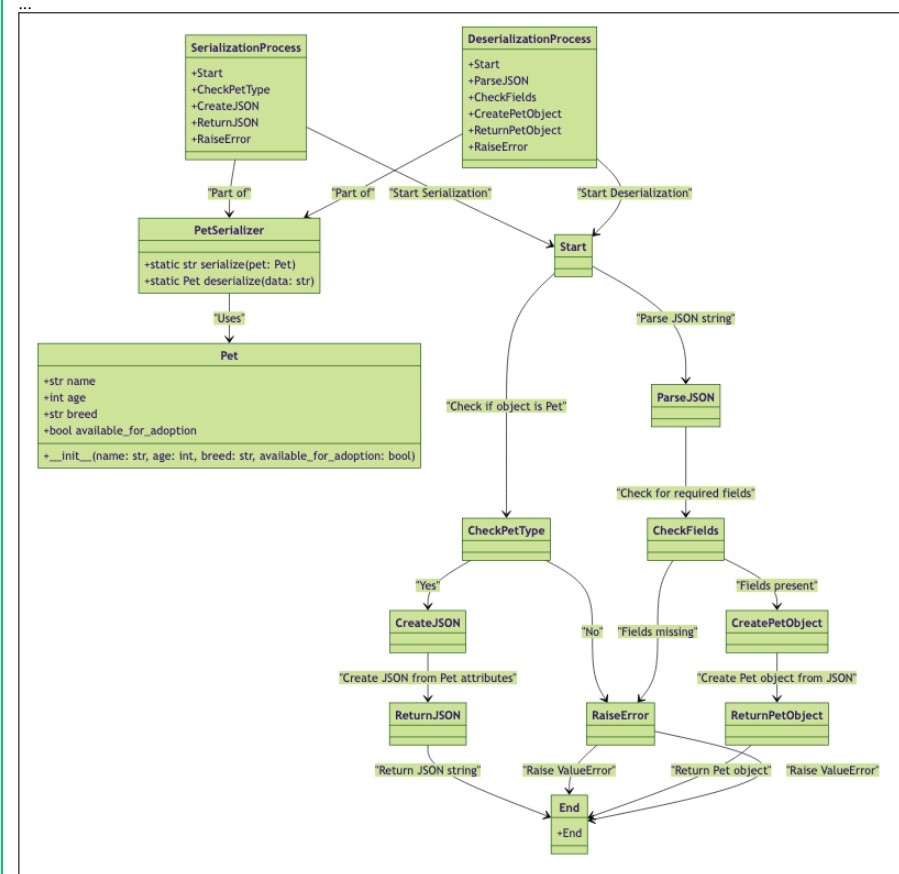

### Response

```python
import json

class Pet:
    def __init__(self, name, age, breed, available_for_adoption):
        self.name = name
        self.age = age
        self.breed = breed
        self.available_for_adoption = available_for_adoption

class PetSerializer:
    @staticmethod
    def serialize(pet):
        # Check if the object is of type Pet
        if isinstance(pet, Pet):
            # Create JSON representation of the pet's attributes
            return json.dumps({
                "name": pet.name,
                "age": pet.age,
                "breed": pet.breed,
                "available_for_adoption": pet.available_for_adoption
            })
        else:
            # Raise ValueError if the object is not of type Pet
            raise ValueError("Object is not of type Pet")
...
```

This solution follows the processes outlined in the diagram, ensuring that the 'PetSerializer' class correctly serializes and deserializes 'Pet' objects while handling errors appropriately.

32

*Figure 26.* Diagram example 5 from M²C-INSTRUCT.

## D.2. M²EVAL Examples

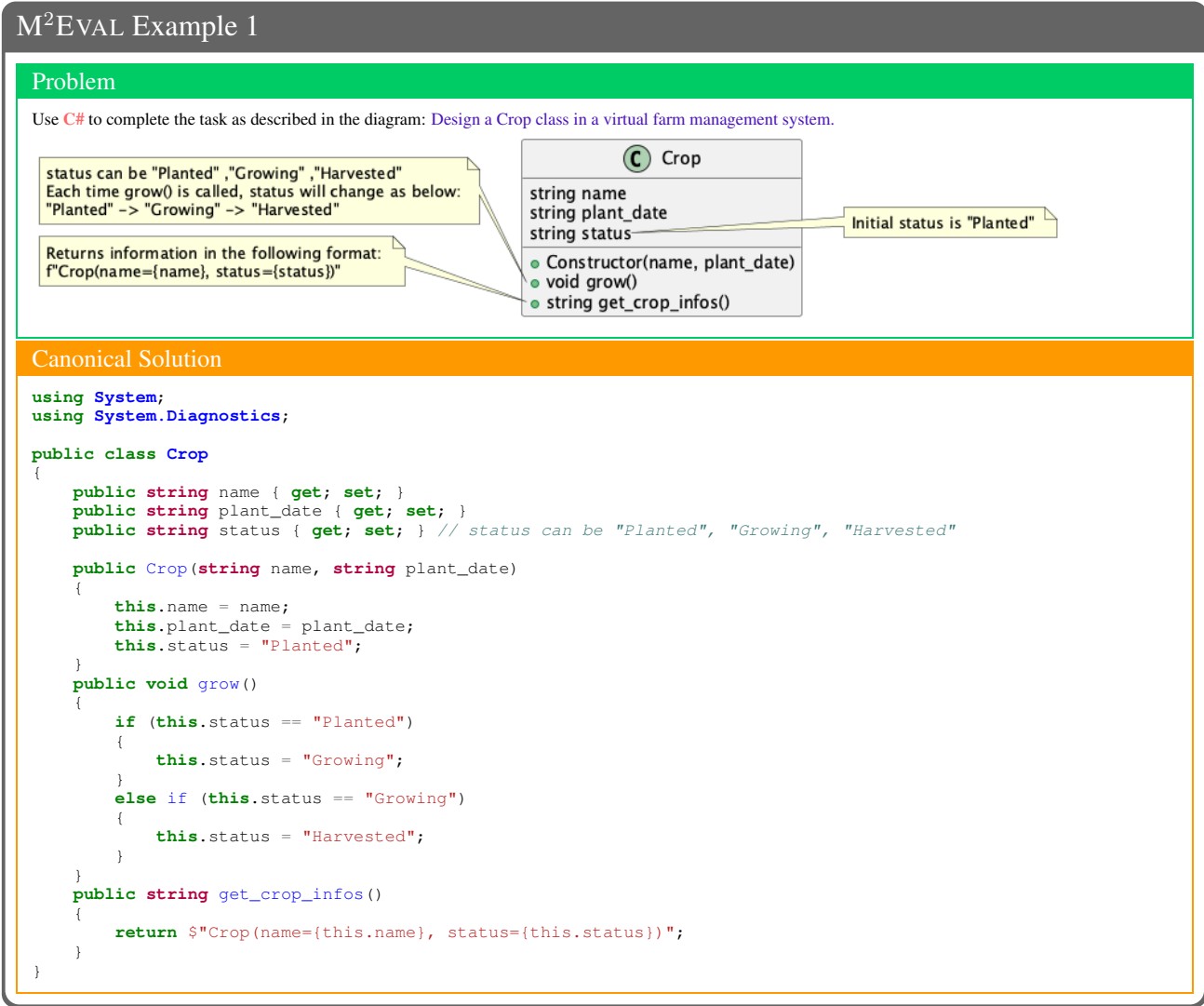

*Figure 27.* Example 1 from M²EVAL.

## M²Eval Example 2

### Problem

Use Python to complete the task as described in the diagram: Design **Crop(abstract), Wheat and Corn class** in a virtual farm management system.

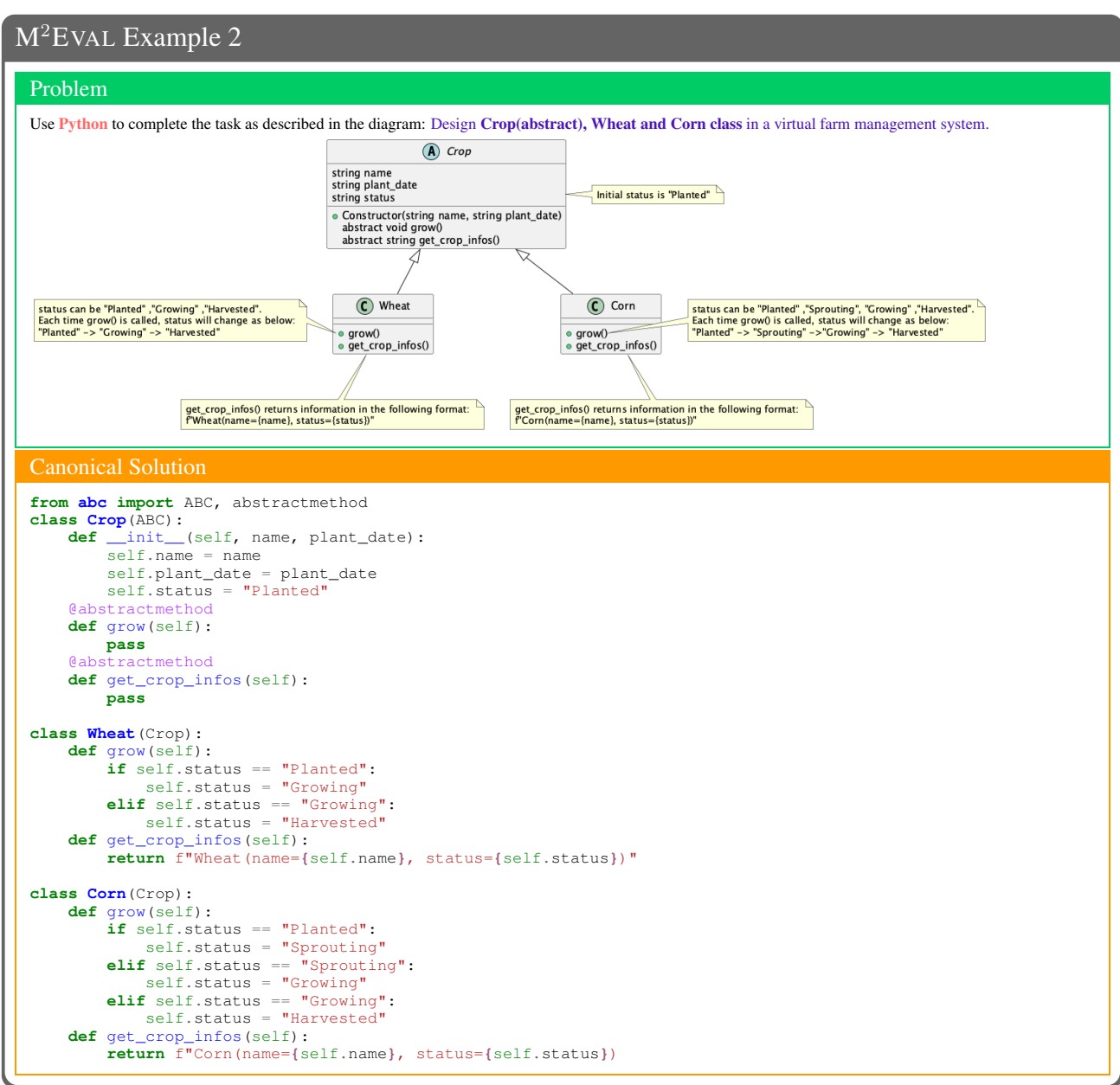

### Canonical Solution

```python
from abc import ABC, abstractmethod
class Crop(ABC):
    def __init__(self, name, plant_date):
        self.name = name
        self.plant_date = plant_date
        self.status = "Planted"
    @abstractmethod
    def grow(self):
        pass
    @abstractmethod
    def get_crop_infos(self):
        pass

class Wheat(Crop):
    def grow(self):
        if self.status == "Planted":
            self.status = "Growing"
        elif self.status == "Growing":
            self.status = "Harvested"
    def get_crop_infos(self):
        return f"Wheat(name={self.name}, status={self.status})"

class Corn(Crop):
    def grow(self):
        if self.status == "Planted":
            self.status = "Sprouting"
        elif self.status == "Sprouting":
            self.status = "Growing"
        elif self.status == "Growing":
            self.status = "Harvested"
    def get_crop_infos(self):
        return f"Corn(name={self.name}, status={self.status})
```

*Figure 28.* Example 2 from M²Eval.

## M²Eval Example 3

### Problem

Use Kotlin to complete the task as described in the diagram: Design GameCharacter(abstract), Warrior, Mage, GameWorld class and a CharacterFactory class to create characters of type Warrior or Mage in a VR game world where users can create characters, explore the world, and interact with other characters.

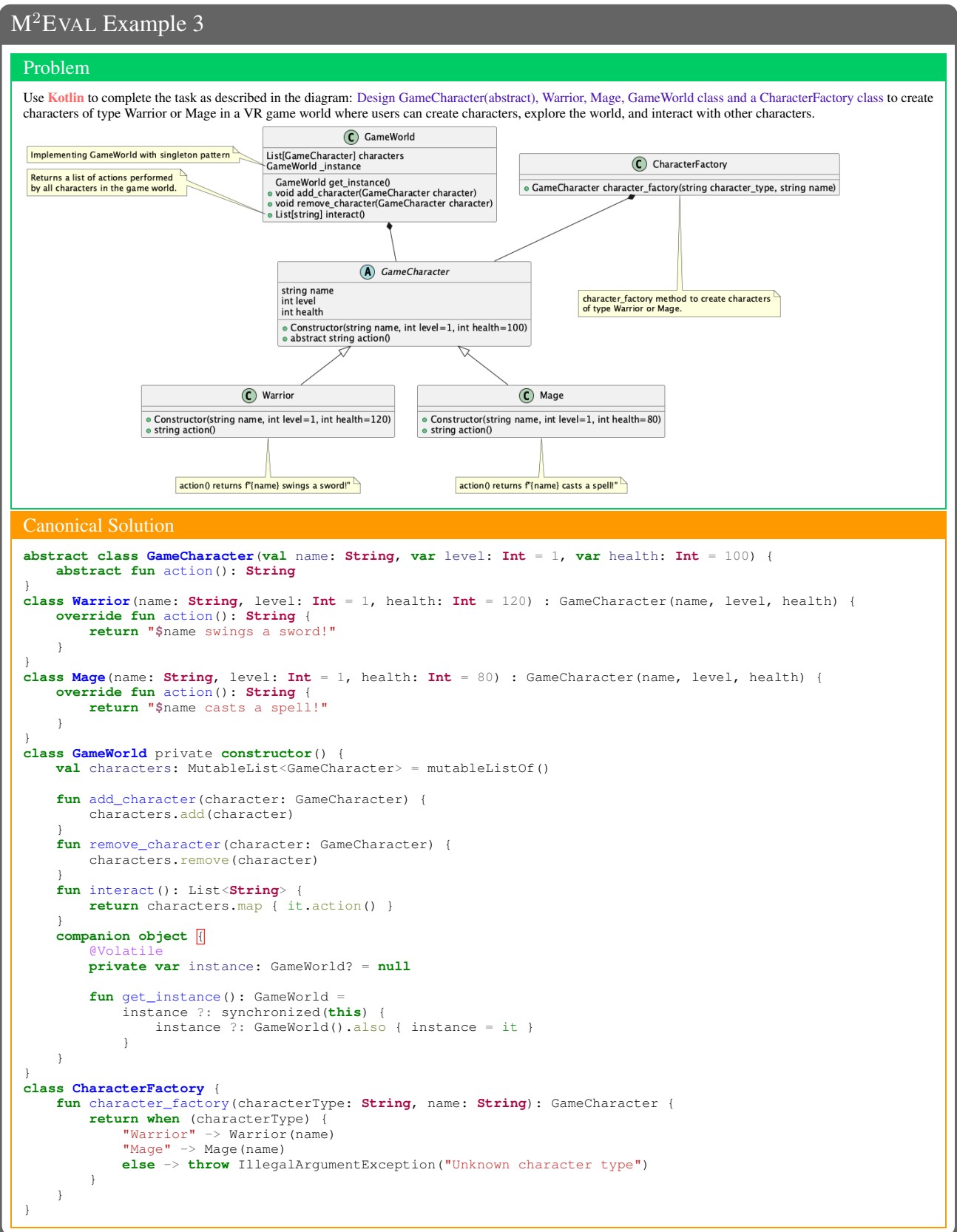

### Canonical Solution

```kotlin
abstract class GameCharacter(val name: String, var level: Int = 1, var health: Int = 100) {
    abstract fun action(): String
}
class Warrior(name: String, level: Int = 1, health: Int = 120) : GameCharacter(name, level, health) {
    override fun action(): String {
        return "$name swings a sword!"
    }
}
class Mage(name: String, level: Int = 1, health: Int = 80) : GameCharacter(name, level, health) {
    override fun action(): String {
        return "$name casts a spell!"
    }
}
class GameWorld private constructor() {
    val characters: MutableList<GameCharacter> = mutableListOf()

    fun add_character(character: GameCharacter) {
        characters.add(character)
    }
    fun remove_character(character: GameCharacter) {
        characters.remove(character)
    }
    fun interact(): List<String> {
        return characters.map { it.action() }
    }
    companion object {
        @Volatile
        private var instance: GameWorld? = null

        fun get_instance(): GameWorld =
            instance ?: synchronized(this) {
                instance ?: GameWorld().also { instance = it }
            }
    }
}
class CharacterFactory {
    fun character_factory(characterType: String, name: String): GameCharacter {
        return when (characterType) {
            "Warrior" -> Warrior(name)
            "Mage" -> Mage(name)
            else -> throw IllegalArgumentException("Unknown character type")
        }
    }
}
```

*Figure 29.* Example 3 from M²Eval.

## M²Eval Example 4

### Problem

Use Use **Swift** to complete the task as described in the diagram: Design an AI assistant system to manage different types of AI assistants and handle various user requests. You need Design AIAssistant(abstract), WeatherAssistant, NewsAssistant and HealthAssistant classes to handle different types of AI assistants, and you need to design a User class to manage different types of AI assistants.

### Canonical Solution

```ruby
class AIAssistant
  attr_reader :name, :version, :capabilities

  def initialize(name, version, capabilities)
    @name = name
    @version = version
    @capabilities = capabilities
  end

  def answer_question(question)
    raise NotImplementedError, "Subclasses must implement the answer_question method"
  end
end

class WeatherAssistant < AIAssistant
  def initialize(name, version)
    super(name, version, ["weather"])
  end

  def answer_question(question)
    if question.downcase.include?("weather")
      "#{@name} says: The weather is sunny!"
    else
      "#{@name} can't answer this question."
    end
  end
end
...
class User
  def initialize
    @assistants = []
  end

  def add_assistant(assistant)
    @assistants << assistant
  end

  def remove_assistant(assistant)
    @assistants.delete(assistant)
  end

  def assign_request(question)
    @assistants.each do |assistant|
      response = assistant.answer_question(question)
      return response unless response.include?("can't answer")
    end
    "None of the assistants can answer this question."
  end
end
```

*Figure 30.* Example 4 from M²Eval.

## M²EVAL Example 5

### Problem

Use **PHP** to complete the task as described in the diagram: Write a function 'function calculate_number(int $number): int ' to calculate the number.

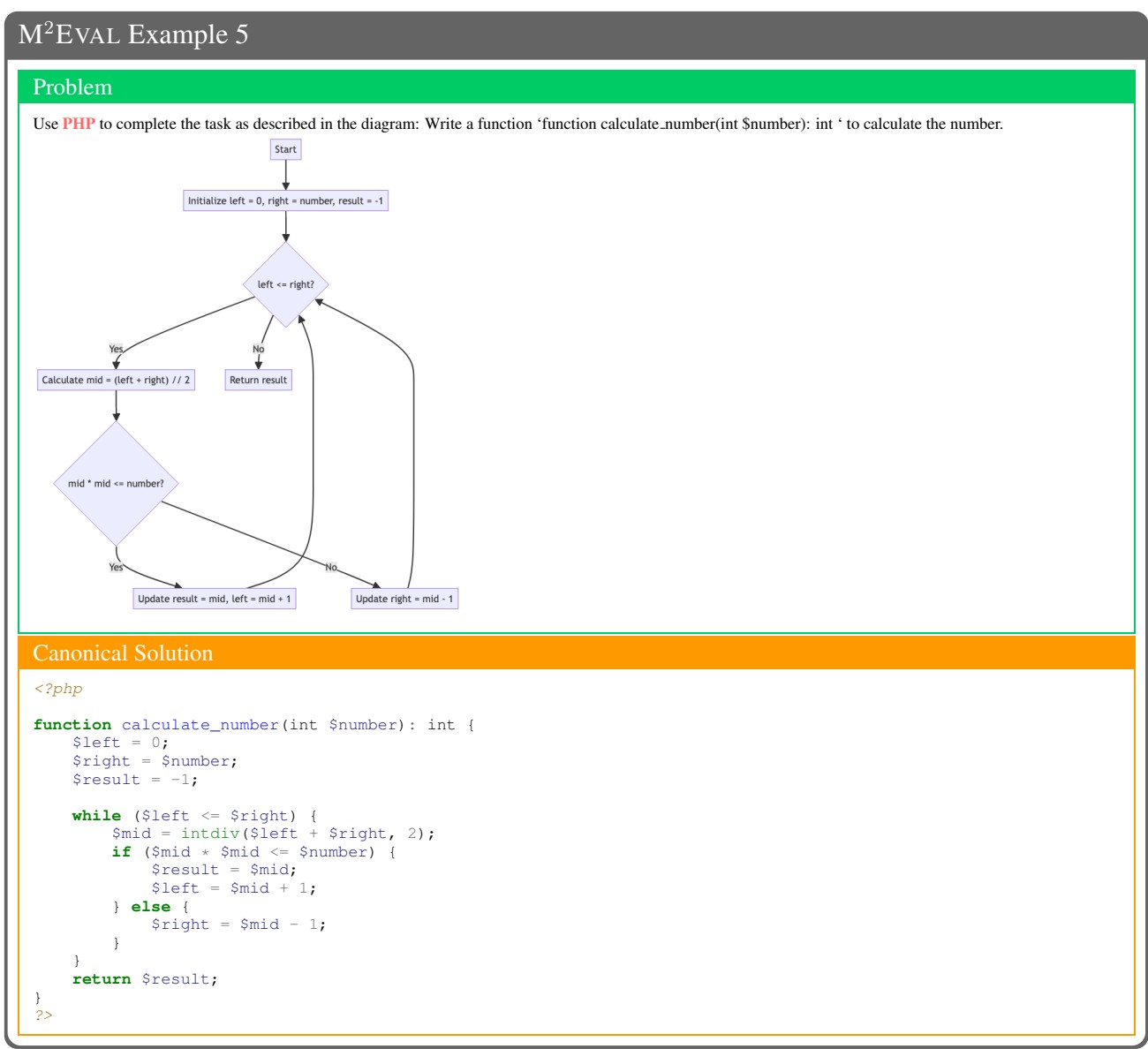

### Canonical Solution

```php
<?php

function calculate_number(int $number): int {
    $left = 0;
    $right = $number;
    $result = -1;

    while ($left <= $right) {
        $mid = intdiv($left + $right, 2);
        if ($mid * $mid <= $number) {
            $result = $mid;
            $left = $mid + 1;
        } else {
            $right = $mid - 1;
        }
    }
    return $result;
}
?>
```

*Figure 31.* Example 5 from M²EVAL.

## M²EVAL Example 6

### Problem

Use **Scala** to complete the task as described in the diagram: Write a function 'def find_word(paragraph: String, banned: List[String]): Option[String]' to find a special word in a paragraph.

```
Start
  │
  ▼
Initialize banned_set with lowercase banned words
  │
  ▼
Extract words from paragraph
  │
  ▼
Filter out banned words
  │
  ▼
Count occurrences of each word
  │
  ▼
Is words less than 2?
  │
Yes ┌──────┴──────┐ No
  ▼                ▼
Return None    Get most common words
                   │
                   ▼
Sort most common words by frequency
in descending order and lexicographical order in descending order
                   │
                   ▼
Return the most frequent word
                   │
                   ▼
                  End
```

### Canonical Solution

```scala
import scala.collection.mutable
import scala.math.Numeric.IntIsIntegral
import scala.math.Ordering.Implicits._
import scala.util.matching.Regex

object Main {
  def find_word(paragraph: String, banned: List[String]): Option[String] = {
    val bannedSet = banned.map(_.toLowerCase).toSet
    val words = new Regex("\\w+").findAllIn(paragraph.toLowerCase).toList
    val filteredWords = words.filterNot(bannedSet.contains)
    val wordCounts = mutable.Map[String, Int]().withDefaultValue(0)

    filteredWords.foreach(word => wordCounts(word) += 1)

    if (wordCounts.size < 2) {
      None
    } else {
      val mostCommon = wordCounts.toList.sortBy { case (word, count) => (-count, word.map(-_.toInt)) }
      Some(mostCommon.head._1)
    }
  }
}
```

*Figure 32.* Example 6 from M²EVAL.

## M²EVAL Example 7

### Problem

Use **JavaScript** to complete the task as described in the diagram: Write a function 'function navigate_complex_road(road_conditions)' to solve the following problem: The function should analyze the sequence of road conditions and decide on the appropriate actions to ensure safe and efficient navigation.
Args: road_conditions (List[str]): A list of strings representing the sequence of road conditions the vehicle will encounter.
Returns: List[str]: A list of strings representing the actions the vehicle should take to navigate through the given road conditions.

### Canonical Solution

```javascript
function navigate_complex_road(road_conditions) {
    const actions = [];
    for (let condition of road_conditions) {
        switch (condition) {
            case "clear":
                actions.push("accelerate");
                break;
            case "construction":
                actions.push("decelerate");
                break;
            case "traffic_jam":
                actions.push("stop");
                break;
            case "turn_left":
                actions.push("turn_left");
                break;
            case "turn_right":
                actions.push("turn_right");
                break;
            default:
                actions.push("unknown");
        }
    }
    return actions;
}
```

*Figure 33.* Example 7 from M²EVAL.

## M²EVAL Example 8

### Problem

Use **CPP** to complete the task as described in the diagram:
Write a function 'unordered_map¡string, vector¡string¿¿ virtual_meeting_assistant(const string& operation, const unordered_map¡string, string¿& data)' that performs different operations based on the given operation type and data. The function should manage the following tasks: - Schedule a Meeting - Add Participants - Record Meeting Minutes - Generate a Summary
Args:
- operation (str): The type of operation to perform. It can be one of the following: ...
- data (dict): A dictionary containing the necessary data for the operation. ...
Return: For "generate_summary", return a dictionary with a key "summary" and a list of key points as the value. For other operations, return an empty dictionary.

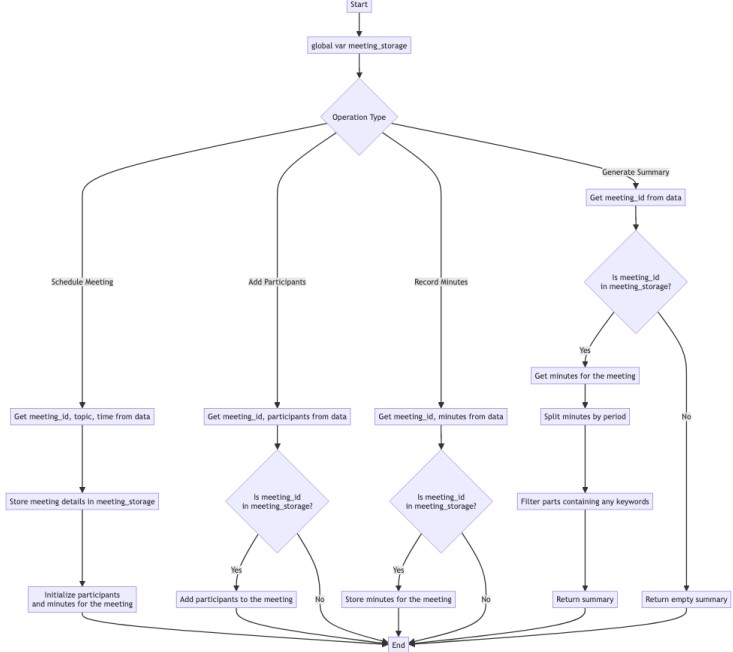

### Canonical Solution

```cpp
#include <iostream>
...
using namespace std;
struct MeetingStorage {
    unordered_map<int, pair<string, string>> meetings;
    unordered_map<int, vector<string>> participants;
    unordered_map<int, string> minutes;
};
MeetingStorage meeting_storage;

unordered_map<string, vector<string>> virtual_meeting_assistant(const string& operation,
const unordered_map<string, string>& data) {
    if (operation == "schedule") {
        ...
    }
    else if (operation == "add_participant") {
        int meeting_id = stoi(data.at("meeting_id"));
        if (meeting_storage.participants.find(meeting_id) != meeting_storage.participants.end()) {
            ...
        }
    }
    else if (operation == "record_minutes") {
        int meeting_id = stoi(data.at("meeting_id"));
        string minutes = data.at("minutes");
        ...
    }
    else if (operation == "generate_summary") {
        int meeting_id = stoi(data.at("meeting_id"));
        ...

        return {{"summary", key_points}};
    }

    return {};
}
```

*Figure 34.* Example 8 from M²EVAL.

# E. Detailed Related Work

**Code Large Language Model.** With the rapid advancement of large language models(LLMs)(OpenAI, 2023; Touvron et al., 2023; Dubey et al., 2024; Bai et al., 2023a; Yang et al., 2024a), solving complex code-related tasks has become increasingly feasible, leading to the emergence of numerous Code LLMs. Early studies utilized models like BERT(Devlin et al., 2019) or GPT(Radford et al., 2018) as backbones, trained on billions of code snippets to enable tasks involving code understanding and generation(Chen et al., 2021; Feng et al., 2020; Scao et al., 2022; Li et al., 2022; Wang et al., 2021; Allal et al., 2023). Recently, advancements in domain-specific pre-training and instruction fine-tuning techniques have led to extensive efforts in fine-tuning models on large-scale code corpora and crafting code-related task instructions(Rozière et al., 2023; Zheng et al., 2023; Luo et al., 2023; Muennighoff et al., 2023; Gemma Team, 2024; Zheng et al., 2024a; Guo et al., 2024; Wei et al., 2024; Sun et al., 2024; Lozhkov et al., 2024; Jiang et al., 2023; Hui et al., 2024). These models demonstrate remarkable performance in tasks like code completion, synthesis, and interpretation.

**Code Evaluation.** To assess the code capabilities of LLMs, a wide range of benchmarks have been developed to evaluate code quality, functionality, and efficiency. Early efforts (Chen et al., 2021; Austin et al., 2021; Liu et al., 2023b; Zhuo et al., 2024) concentrated on the fundamental coding abilities of LLMs. However, given the complexity and multifaceted nature of code tasks, subsequent evaluations have expanded to include code repair (Lin et al., 2017; Tian et al., 2024; Zhang et al., 2023; Prenner & Robbes, 2023; He et al., 2022; Wang et al., 2023a), multilingual code assessments (Zheng et al., 2023; Cassano et al., 2023; Wang et al., 2023b; Athiwaratkun et al., 2023; Chai et al., 2024), repository-level evaluations (Liu et al., 2023c; Deng et al., 2024; Bairi et al., 2024), agent-based code evaluations (Jimenez et al., 2023; Zhang et al., 2024e), and more.

**Visual Reasoning and Code Synthesis.** Large Language Multmodal Models (LMMs)(Zhu et al., 2023; Liu et al., 2023a; Bai et al., 2023b; Ye et al., 2023; Liu et al., 2024b; Zhang et al., 2024b; Li et al., 2024a; Wang et al., 2024a; Zong et al., 2024) incorporate visual information into LLMs through visual encoders(Radford et al., 2021), extending the capabilities of LLMs to address visual tasks. Prior studies such as VQA(Antol et al., 2015; Goyal et al., 2017; Tang et al., 2024) evaluated the basic visual semantic capabilities of models. With the emergence of increasingly powerful visual and semantic capabilities in LMMs and LLMs, many recent works have shifted focus toward more complex multi-modal tasks, such as mathematical reasoning(Trinh et al., 2024; Shao et al., 2024; Huang et al., 2024; Zhang et al., 2024c), chart understanding(Han et al., 2023; Tannert et al., 2023; Singh et al., 2024; Wang et al., 2024b; Li et al., 2024c; Zhang et al., 2024d), code generation(Si et al., 2025; Li et al., 2024b; Liu et al., 2025; Wu et al., 2024; Shi et al., 2024; Mu et al., 2024), and agent-driven interactions(Zhou et al., 2024; Xie et al., 2024; Liu et al., 2024c).

Recent multimodal code works focused on visual algorithmic problems(Li et al., 2024b; Wang et al., 2025), chart code generation(Wu et al., 2024; Shi et al., 2024), and UI design(Si et al., 2025; Liu et al., 2025). Our M$^2$EVAL explores the task of code generation based on code diagrams. Compared with previous work(Liu et al., 2022) that converts flowcharts into node information for complex processing, we focus on more practical scenarios that directly perform semantic understanding based on images.

