# OpenReview forum: "From Diagrams to Code: Multilingual Programming with Visual Design"
_ICML.cc/2026/Conference — ICML 2026 regular_

### Official Review · Reviewer_ore8 · 2026-02-24

**Soundness:** 3
**Presentation:** 3
**Significance:** 3
**Originality:** 2
**Overall Recommendation:** 4
**Confidence:** 3

**Summary:**

This paper targets emerging “vibe coding” and visual-first development workflows by enabling code generation directly from visual design artifacts such as UML diagrams and flowcharts. Unlike prior multimodal code-generation work that focuses on narrow domains (e.g., chart-to-code or UI-to-HTML) and treats visuals as auxiliary hints, the paper positions diagrams as the primary specification and addresses the full software design workflow.

The authors introduce M2C-INSTRUCT, a large multilingual multimodal instruction-tuning dataset with 13.1M examples spanning 50+ programming languages, containing both cross-modal code-understanding tasks and diagram-centric problems where key information is only available visually. Using this dataset, they train M2-CODER, a 7B-parameter multimodal model that takes an image and textual instructions to generate executable code via a two-stage fine-tuning procedure. They also propose M2EVAL, a human-curated benchmark covering 10 programming languages and 30 programming concepts for evaluating diagram-to-code generation using execution-based Pass@1.

Experiments over 25+ models show that M2-CODER (7B) achieves Pass@1 performance comparable to much larger 70B+ LMMs, and that strong text-only code LLMs obtain 0% accuracy when diagrams are removed, validating the necessity of visual inputs. Overall, the paper contributes a large-scale multimodal dataset, a trained multimodal code model, and a new benchmark for visual-first multilingual programming.

**Compliance With Llm Reviewing Policy:**

Affirmed.

**Final Justification:**

The authors addressed my concern in the rebuttal.

**Key Questions For Authors:**

1. How were the 30 problem concepts in M²EVAL selected, and what evidence supports that they are representative of the broader diagram-to-code task space?
2. What is the Mermaid rendering success rate during dataset synthesis? How many diagrams were discarded due to rendering failures, and could this introduce selection bias toward simpler or more formulaic diagram structures?
3. Could you provide confidence intervals or statistical significance tests for Pass@1 results, given the small number of problems per language (30 per language, binary pass/fail)?
4. Qwen3-VL models were publicly available several months before the submission deadline. Why was Qwen2-VL chosen as the base for M²-CODER? Have you considered or experimented with fine-tuning Qwen3-VL-base, and would this change the comparative conclusions regarding the effectiveness of M²C-INSTRUCT?

**Limitations:**

The paper would benefit from an explicit limitations section. In particular, important limitations are not sufficiently discussed:

(1) the small size and narrow scope of M²EVAL (30 concepts * 6 languages), which constrains statistical power and coverage;
(2) potential noise and distributional bias in large-scale synthetic instruction data generated via LLM pipelines; and
(3) the restriction to a small set of diagram types (class, sequence, flowchart, and state).

A dedicated discussion of these factors would improve transparency and help contextualize the reported gains.

**Strengths And Weaknesses:**

Strengths:
- Well-motivated and practical problem setting. The “diagrams to code” framing addresses a genuine gap in current multimodal code generation. In practice, software architects communicate designs via UML and flowcharts, and the paper correctly observes that existing multimodal code benchmarks are largely monolingual and domain-restricted. The comparison in Table 3 effectively highlights this gap.
- Comprehensive benchmark design. M²EVAL is carefully constructed through a three-step process (prototype design → multimodal conversion → multilingual translation) with human curation. The design choice to make diagrams necessary by removing redundant textual information is principled and is validated by the 0% performance of strong text-only code LLMs (DeepSeek-V3, Qwen2.5-Coder) in Table 4.
- Strong empirical results relative to model size. M²-CODER-7B achieving 25.3% average Pass@1, outperforming Qwen2.5-VL-72B (24.7%) and approaching proprietary models, provides convincing evidence of the effectiveness of the proposed training data. The ablation study (Table 5) clearly demonstrates the contribution of each training stage.
- Scale and breadth of training data. The 13.1M-sample dataset spanning 50+ languages with 42.3M images reflects substantial engineering effort. The combination of cross-modal OCR-style problems and diagram-centric problems is well motivated.

Weaknesses:
- M²EVAL contains only 30 unique problems (300 total with translations), which limits coverage and statistical power. With only 30 concepts, diversity of real-world diagram-to-code scenarios is likely underrepresented, and per-category analyses are based on very small sample sizes.
- Despite broad “visual design” framing, the benchmark uses only UML class diagrams and flowcharts. Many common diagram types (sequence, state, ER, architecture, wireframes) are not considered.
- The 13.1M-sample M²C-INSTRUCT dataset is fully LLM-synthesized, yet no systematic quality analysis (execution filtering, human audit, or correctness rate) is reported, raising concerns about noise and compounding errors.
- Results rely solely on greedy Pass@1 over small test sets (30 problems per language). Confidence intervals or statistical tests are absent, making small performance differences difficult to interpret.
- M²-CODER uses a standard Qwen2-VL backbone with two-stage SFT and introduces no architectural or objective-level innovations; the primary contribution is the dataset.
- Important ablations are missing (e.g., cross-modal vs diagram-only training, data ratio choices, lightweight vs full Stage-1 data).
- Recent strong multimodal baselines from the same family (e.g., Qwen3-VL series) are not evaluated, which weakens comparative claims.

---

> ### Author Rebuttal · Authors · 2026-03-31
>
> We sincerely thank you for your positive evaluation and recognition of the dataset scale, benchmark rigor, and language diversity. We address each concern below.
>
> ---
>
> **Benchmark size, concept selection, and statistical power**
>
> Rather than pursuing scale alone, we prioritize high-quality human curation paired with detailed error analysis (Section 4.3, Figure 7) to surface concrete failure modes—actionable insights that a larger but noisier benchmark would obscure.
>
> M2EVAL comprises 30 concepts × 10 languages = 300 instances. Bootstrap confidence intervals (B=10,000) yield a 95% CI of [21.0, 30.0], and McNemar tests confirm significance vs. all 7B-8B baselines (p<0.001) but not vs. 72B models (p=0.26--0.90). The 30 concepts span 4 categories (Table 6) with 9 Easy / 7 Medium / 14 Hard problems. Since M2EVAL targets function- and class-level code generation, the relevant diagram types are inherently limited to UML class diagrams and flowcharts. HumanEval (164 problems) similarly uses a curated subset yet remains foundational. All problems are human-designed via our three-step process (Section 2.2) with cross-verification and a 100% test-pass rate, and diagram necessity is validated by 0% accuracy from text-only LLMs. In future work, we plan to extend to repository-level scenarios involving more diverse diagram types (sequence, architecture, state diagrams).
>
> ---
>
> **Training data quality and noise**
>
> Recent work has consistently shown that large-scale synthetic data emphasizing diversity, in mid-training or first-stage SFT, can significantly boost code, agent, and visual capabilities (e.g., Qwen3, GLM-5). Our Stage-1 follows this principle with 12.9M samples across 50+ languages. A human audit of 500 samples (Fleiss' {\displaystyle \kappa }=0.79) shows 92.4% code correctness, 88.6% diagram-code consistency, and 94.6% instruction clarity. Our multi-stage quality control pipeline (Section 3.2, Appendix B.2) applies StarCoder filtering, followed by three-step diagram synthesis with Mermaid rendering validation (~12% filtered), and prompt constraints ensuring key information resides only in diagrams. Overall, 18--22% of candidates are removed, and M2-CODER's strong results confirm a sufficient signal-to-noise ratio.
>
> ---
>
> **Mermaid rendering success rate and selection bias**
>
> The rendering success rate is 87.9% (1.70M of 1.93M). Failed diagrams are significantly more complex (12.4 vs. 7.8 nodes, p<0.001), confirming a mild bias toward moderate complexity. The retained set still spans 2--48 nodes with heights 37--8,018px (Table 9). We will discuss this in the Limitations section.
>
> ---
>
> **Data composition ablations**
>
> Beyond Table 5 and Figure 6, we provide a finer-grained Stage-1 data ablation (Table R9):
>
> | Stage-1 Data | Pass@1 |
> |:---|:---:|
> | 0% (Stage-2 only) | 18.0% |
> | 25% | 18.3% |
> | 50% | 21.0% |
> | 75% | 23.0% |
> | 100% | 25.3% |
>
> Performance has not plateaued at 100%, suggesting room for further scaling. Combined with Table 5 (Stage-1 only: 10.0%), the two stages are clearly complementary. Table 5 also ablates cross-modal vs. diagram-only components, confirming each contributes distinct capabilities (e.g., removing Stage-1 cross-modal data drops Pass@1 from 25.3% to 18.0%).
>
> ---
>
> **Dataset-centric contribution**
>
> We acknowledge that M2-CODER employs standard two-stage SFT without architectural modification, and the primary contribution is data-centric. This is by design: foundational works (Alpaca, WizardCoder, Magicoder, LLaVA-Instruct) have driven advances through data quality and scale rather than architectural novelty. M2C-INSTRUCT fills a previously unoccupied niche—large-scale multimodal code instruction data spanning 50+ languages—and our results show that data alone enables a 7B model to match 72B alternatives. The dataset and benchmark will be fully released to support future architectural innovations in this space.
>
> ---
>
> **Qwen3-VL baselines and base model choice**
>
> To validate cross-model generality, we fine-tuned Qwen3-VL-8B-Instruct with our full two-stage pipeline (Table R7), yielding a +8.3 point gain over Qwen3-8B zero-shot.
>
> | Model | C# | CPP | Java | JS | Kotlin | PHP | Python | Ruby | Scala | Swift | Avg |
> |:---|:---:|:---:|:---:|:---:|:---:|:---:|:---:|:---:|:---:|:---:|:---:|
> | Qwen3-8B | 26.7 | 26.7 | 26.7 | 30.0 | 23.3 | 33.3 | 33.3 | 30.0 | 20.0 | 16.7 | 26.7 |
> | **M2-Qwen3** | **36.7** | **33.3** | **36.7** | **40.0** | **30.0** | **43.3** | **46.7** | **40.0** | **23.3** | **20.0** | **35.0** |
> | Qwen3-32B | 36.7 | 36.7 | 40.0 | 43.3 | 33.3 | 40.0 | 46.7 | 36.7 | 26.7 | 23.3 | 36.3 |
>
> The fine-tuned 8B nearly matches the 4× larger Qwen3-32B overall, confirming that task-specific SFT can effectively compensate for model scale.
>
> ---
>
> We appreciate your thorough evaluation. All requested experiments and a dedicated Limitations section, covering benchmark scope and statistical power, synthetic data bias, and diagram type restrictions, will be included in the next version.

---

> > ### Author Rebuttal · Reviewer_ore8 · 2026-04-03
> >
> > The authors adequately addressed my main concerns. The authors appear to analyze the problem from a practical and well-motivated angle, and the authors explore a central context that fills a genuine gap in multimodal code generation. The new statistical tests, human audit results, Qwen3-VL experiments, and data scaling ablations are convincing. I raise my score to Weak Accept, assuming the camera-ready incorporates the promised additions (confidence intervals, Qwen3 results, Limitations section, and tempered 70B+ claims).

---

### Official Review · Reviewer_bbeB · 2026-02-28

**Soundness:** 3
**Presentation:** 3
**Significance:** 2
**Originality:** 4
**Overall Recommendation:** 5
**Confidence:** 4

**Summary:**

They introduce a comprehensive framework designed to bridge the gap between visual software architecture and executable code. They argue that people first think with diagrams and visual interactions for tasks like system designs before writing lines of code but the current language models are not capable of converting these kinds of diagrams, like UML or flowcharts to code. So they propose an instruct dataset for training with 13 million samples, a 7B model based on qwen2-VL trained on their dataset, and a benchmark consisting of 300 problems across 10 programming languages.

**Compliance With Llm Reviewing Policy:**

Affirmed.

**Key Questions For Authors:**

1. How do you mitigate "hallucinated" logic during the Step 2 Diagram Synthesis?
2. To what extent did the 12.9M Stage-1 samples contain "noisy" syntax highlighting? You used Pygments to render code into images; however, did you account for the wide variety of IDE themes and font styles developers actually use, or is the model overfitted to a specific visual "look"?
3. Does the model support "Image-to-Image" iterative refinement? In a standard developer workflow, design is an iterative process. If a developer modifies a diagram slightly, can the model perform "delta-updates" to the code, or does it have to regenerate the entire file from scratch, potentially losing previous manual edits?

**Strengths And Weaknesses:**

#### Strengths:
- The creation of $M^2C-INSTRUCT$ is a major contribution spanning over 50 programming languages
- The benchmark is human-curated, which makes it reliable compared to synthetic benchmarks.
- comprehensive ablation study showing their two-stage training brings improvement while being efficient to some extent

#### Weaknesses:
- Training a 7B model may limit its capabilities to learn this task, and also small models are weaker in other languages (other than Python and C).
- Data generation is heavily based on LLMs
- While the benchmark is human-curated, the diagrams are still largely synthetic or idealized. Real-world designs are often messy and hand-drawn. The real designs are hardly as clean as the Mermaid diagrams.

---

> ### Author Rebuttal · Authors · 2026-03-31
>
> We sincerely thank you for your positive evaluation and thoughtful questions on data quality, visual diversity, and iterative refinement. Below we address each concern.
>
> ---
>
> **7B model capacity and non-Python language performance**
>
> We agree that 7B models face capacity limitations for lower-resource languages. Per-language performance ranges from 16.7% (Kotlin, Scala, Swift) to 36.7% (PHP, Python). Data scale ablation (Table R9):
>
> | Stage-1 Data | Pass@1 |
> |:---|:---:|
> | 0% (Stage-2 only) | 18.0% |
> | 25% | 18.3% |
> | 50% | 21.0% |
> | 75% | 23.0% |
> | 100% | 25.3% |
>
> Performance has not plateaued, suggesting further scaling could improve weaker languages. Our core contribution is the dataset and benchmark; fine-tuning larger models on M2C-INSTRUCT is a natural extension. We chose 7B to demonstrate that high-quality training data can compensate for model scale.
>
> ---
>
> **LLM-based data quality and hallucination mitigation**
>
> We acknowledge LLM-based synthesis as a core pipeline aspect. Recent work has consistently demonstrated that large-scale synthetic data emphasizing diversity—typically used in mid-training or first-stage SFT—can significantly boost code, agent, and visual capabilities (e.g., Qwen3, GLM-5). Our Stage-1 follows this principle with 12.9M diverse samples across 50+ languages. We have quantified the resulting data quality: human evaluation of 500 samples yields 92.4% code correctness and 88.6% diagram-code consistency. Multi-stage filtering removes 18--22% of candidates, retaining 12.9M Stage-1 and 168K Stage-2 samples.
>
> Diagram hallucination audit (Table R10, 200 samples):
>
> | Type | Rate |
> |:---|:---:|
> | Overall hallucination | 9.0% |
> | — Missing class attributes | 4.5% |
> | — Extra methods | 3.5% |
> | — Incorrect relationships | 3.0% |
> | High-severity only | 3.0% |
>
> Categories overlap, so type rates sum above 9.0%. Only 3.0% are high-severity (fabricated classes or incorrect control flow). Our pipeline mitigates hallucinations via the three-step synthesis methodology (Appendix B.2): (1) Qwen2.5-Coder-32B generates Mermaid diagrams, with unsuccessful renderings filtered out (~12% removed); (2) the prompt template (Figure 14) explicitly instructs "ensure that vital information, indispensable for solving the coding challenge, is exclusively contained within the diagram," enforcing alignment by design; (3) final mermaid-cli rendering and post-hoc complexity filtering remove additional outliers. Source data is preprocessed via the StarCoder pipeline (Appendix B.2) with rule-based filtering before any LLM synthesis. This noise level is acceptable for training, validated by M2-CODER's strong performance (7B matching 72B models).
>
> ---
>
> **Gap between synthetic Mermaid diagrams and real-world designs**
>
> We agree that clean Mermaid diagrams differ from messy, hand-drawn diagrams. We view this as analogous to OCR progressing from printed to handwritten text: strong performance on machine-rendered diagrams is a necessary first step. Stage-1's 12.9M cross-modal samples provide visual robustness through content diversity: as documented in Appendix B.1 (Tables 7--9), images span heights 24--13,345px and widths 10--57,246px across 50+ languages. Modern software teams increasingly use digital tools (Lucidchart, draw.io, Mermaid) producing clean diagrams or screen captures similar to our benchmark. Augmenting with synthetic noise (rotation, distortion, annotation artifacts) to bridge the clean-to-messy gap is an important future direction we will discuss in the limitations section.
>
> ---
>
> **Visual diversity of code images**
>
> We took this into consideration when constructing our dataset. Pygments supports various styles of code rendering and font selections. We systematically tried different combinations of rendering styles and fonts to ensure that the fonts could be displayed correctly, filtering out style combinations that exhibited issues such as garbled characters, color overlapping, and text overlap. During data synthesis, we randomly selected from these validated styles, along with randomized code spacing and line number display, to simulate code images encountered in real-world scenarios. Furthermore, we also took this into account when synthesizing diagrams, selecting diverse combinations of styles and colors.
>
> Examples of various styles we have showcased can be seen in Figures 11 and 12 of the Appendix.
>
> ---
>
> **Iterative refinement and delta-update support**
>
> M2-CODER currently operates as a one-shot model without delta-update support. Supporting this would require conditioning on modified diagrams and prior code, plus training on paired revision sequences. A practical interim is placing M2-CODER in an agentic pipeline with diff-based merging. We consider this a compelling future direction.
>
> ---
>
> We appreciate your evaluation. All requested analyses will be reflected in the next version.

---

> > ### Author Rebuttal · Reviewer_bbeB · 2026-04-05
> >
> > I'd like to keep my positive score.

---

### Official Review · Reviewer_8Sc9 · 2026-03-02

**Soundness:** 3
**Presentation:** 3
**Significance:** 2
**Originality:** 3
**Overall Recommendation:** 4
**Confidence:** 4

**Summary:**

This paper addresses the gap in "visual-first" software development, where system architectures are communicated via diagrams before coding. The authors introduce $M^2C$-INSTRUCT, a massive multilingual multimodal dataset with 13.1M samples across 50+ programming languages. They develop $M^2$-CODER, a 7B parameter model trained via a two-stage strategy to integrate visual design inputs with textual instructions. Finally, they present $M^2EVAL$, a human-curated benchmark covering 10 programming languages and "Visual Workflow" tasks like UML and flowcharts. Experimental results show that the 7B $M^2$-CODER matches or outperforms models over ten times its size (70B+) on multimodal tasks.

**Compliance With Llm Reviewing Policy:**

Affirmed.

**Final Justification:**

The authors’ rebuttal has addressed my main concerns. Their clarifications and additional explanations alleviate my earlier doubts to a satisfactory extent. I therefore maintain my positive assessment of the paper.

**Key Questions For Authors:**

1: Modality Priority: In cases where textual instructions and visual diagrams provide conflicting information, how does $M^2$-CODER resolve the discrepancy?

**Limitations:**

See weakness.

**Strengths And Weaknesses:**

Strengths：

1: Massive Dataset Scale: The introduction of 13.1M instruction-tuning samples covering 50+ languages is a significant contribution to the open-source community.

2: Model Efficiency: Demonstrating that a 7B model can achieve performance comparable to 70B+ models like Qwen2-VL-72B validates the high quality of the proposed $M^2C$-INSTRUCT dataset。

3: Rigorous Benchmark Design: $M^2EVAL$ is not merely synthetic; it is human-curated and manually refined to ensure that diagrams are necessary for solving the problems.

4: Language Diversity: Unlike previous works that are mostly monolingual or focus on narrow domains, this work systematically evaluates 10 different programming languages.

Weakness:

1: Limited Concept Breadth: While $M^2EVAL$ contains 300 problems, it only covers 30 unique programming concepts (replicated across 10 languages), which may not fully capture the vast complexity of real-world software engineering

2: Synthetic Data Bias: Much of the 13.1M training data is generated using LLMs, which might inherit systemic biases or limitations from those base models

---

> ### Author Rebuttal · Authors · 2026-03-31
>
> We sincerely thank you for your positive evaluation and recognition of the dataset scale, benchmark rigor, and language diversity. We address each concern below.
>
> ---
>
> **Limited concept breadth in M2EVAL benchmark**
>
> We agree that 30 concepts represent an initial scope. Rather than pursuing scale alone, we prioritize high-quality human-curated problems paired with detailed error analysis (Section 4.3, Figure 7): our case studies reveal concrete failure modes such as missing functions, incorrect attribute visibility, and diagram-text inconsistencies, providing actionable insights into current models' instruction-following and visual hallucination challenges that a larger but noisier benchmark would obscure.
>
> The selection was systematic: 4 categories (Table 6)—Class Design (6), Design Patterns (9), Algorithm (7), Simulation (8)—covering multiple Knowledge Areas, with difficulty levels 9 Easy / 7 Medium / 14 Hard. Importantly, M2EVAL targets function- and class-level code generation, where the relevant diagram types are inherently limited (primarily UML class diagrams and flowcharts). At this granularity, 30 concepts × 10 languages = 300 instances provide sufficient coverage to evaluate models' core capabilities in visual understanding and code generation. For comparison, HumanEval (164 problems) also focuses on a concept subset yet remains the most widely used code generation benchmark. Each problem undergoes rigorous human curation through our three-step process (Section 2.2): two annotators collaboratively design prototypes, manually refine LLM-generated diagrams, and remove redundant text to ensure the diagram is strictly necessary. A cross-checking protocol (Appendix A.3.4) requires each annotator to verify two peers' work, with all canonical solutions achieving a 100% test-case pass rate. Diagram necessity is validated by 0% accuracy from text-only LLMs (Table 4). In future work, we plan to extend M2EVAL to repository-level scenarios, which naturally involve more diverse and abstract diagram types (e.g., sequence diagrams, architecture diagrams, state machines), thereby expanding both concept coverage and diagram diversity.
>
> ---
>
> **Synthetic data bias from LLM-generated training data**
>
> This is a concern shared across the synthetic data paradigm. Recent work has consistently shown that large-scale synthetic data emphasizing diversity—typically used in mid-training or first-stage SFT—can significantly boost code, agent, and visual capabilities (e.g., Qwen3, GLM-5). Our Stage-1 follows this principle with 12.9M diverse samples across 50+ languages. Our quality audit (Table R4a, 500 samples, 3 annotators):
>
> | Metric | Rate |
> |:---|:---:|
> | Code correctness | 92.4% |
> | Diagram-code consistency | 88.6% |
> | Instruction clarity | 94.6% |
>
> Common errors: logic inconsistencies (4.5--5.3%) and missing diagram elements (3.3--5.0%). Our pipeline targets bias through multiple design choices (Section 3.2, Appendix B.2): (1) Stage-1 source data from GitHub is preprocessed via the StarCoder pipeline with rule-based filtering of duplicates, garbled text, and illegal information; (2) Stage-2 draws from two established datasets: Evol-CodeAlpaca and OSS-Instruct (following Magicoder), diversifying coding patterns; (3) diagram synthesis uses a three-step methodology: Qwen2.5-Coder generates Mermaid diagrams with unsuccessful renderings filtered out, a second LLM pass formulates multimodal problems ensuring key info resides only in the diagram, then final rendering with mermaid-cli. This multi-stage filtering removes 18--22% of candidates (87.9% Mermaid rendering success rate). We acknowledge LLM-synthesized data inherently carries some distributional bias. The high quality rates and M2-CODER's strong performance (7B matching 72B models) confirm sufficient training signal quality. We will add a discussion of synthetic data limitations in the revised manuscript.
>
> ---
>
> **Handling conflicts between textual and visual inputs**
>
> We constructed 10 controlled conflict test cases (e.g., text says "implement Singleton" but diagram shows Factory Method). Results (Table R8b):
>
> | Behavior | Rate |
> |:---|:---:|
> | Follows diagram | 80% |
> | Follows text | 10% |
> | Mixed output | 10% |
>
> While based on a limited sample, this visual-first tendency is consistent with our training design: as described in Section 3.2 and Figure 14, M2C-INSTRUCT is constructed so that "key information necessary for solving the coding problem is preserved only within the diagram," training the model to treat visual input as authoritative. In standard M2EVAL, text and diagram are always consistent, so this does not affect benchmark results. We recommend the diagram as authoritative specification for deployment, and plan a larger-scale conflict study in future work.
>
> ---
>
> We sincerely appreciate your evaluation. All requested analyses and expanded discussions will be addressed in the next version.

---

> > ### Author Rebuttal · Reviewer_8Sc9 · 2026-04-01
> >
> > The authors have addressed my major concerns, and I maintain my positive score.

---

### Official Review · Reviewer_aFcx · 2026-03-14

**Soundness:** 3
**Presentation:** 4
**Significance:** 3
**Originality:** 3
**Overall Recommendation:** 5
**Confidence:** 4

**Summary:**

This work introduces a comprehensive suite of resources for the diagram-to-code generation task. Specifically, it presents M2C-INSTRUCT, a large-scale training dataset comprising 13.1M instances and covering 50+ programming languages; M2EVAL, a newly proposed multimodal evaluation benchmark designed to systematically assess multimodal code generation capabilities; and M2-CODER, a 7B-parameter model trained on M2C-INSTRUCT. Experimental results show that M2-CODER achieves performance on M2EVAL that is comparable to models with 70B+ parameters. In addition, the paper provides extensive empirical evaluations and in-depth analyses, offering valuable insights and promising directions for future research in vision-driven programming.

**Compliance With Llm Reviewing Policy:**

Affirmed.

**Key Questions For Authors:**

1.How did you control for contamination between M²C-INSTRUCT and M²EVAL? Please provide quantitative leakage checks.
2.Have you compared M2-CODER with models fine-tuned on purely text-based datasets on the M2EVAL benchmark to demonstrate the necessity of multimodal training data?
3.The generated code in the paper appears to be mainly function-level. Have you evaluated your approach in repository-level diagram-to-code scenarios?

**Limitations:**

yes

**Strengths And Weaknesses:**

Strengths
1. The paper provides a detailed description of the data collection pipeline and human validation process to ensure data quality. The proposed datasets are evaluated across multiple models, including both open-source and proprietary systems of varying scales, which helps strengthen the reliability of the experimental findings. Furthermore, the study includes comprehensive analyses that substantiate the reported results, demonstrating a technically sound experimental design.
Presentation:
2. The paper is clearly written, with thorough explanations of the methodology and dataset construction. The use of notations, figures, and tables is generally consistent and well-structured, which facilitates understanding of the proposed framework and evaluation results.
Significance:
3. The work provides a systematic evaluation of several mainstream models on the task of generating code from visual representations, accompanied by detailed empirical insights and analyses. These findings contribute valuable understanding of current model capabilities and limitations in this emerging area.
4. The proposed M2EVAL benchmark represents a multilingual, human-verified, high-quality benchmark for diagram-to-code generation. Unlike many existing datasets that treat images merely as auxiliary context while relying primarily on textual descriptions, M2EVAL directly evaluates the multimodal capability of models to generate code from visual inputs. In addition, the release of the large-scale M2C-INSTRUCT training dataset provides important resources for future research in multimodal code generation and related areas.

Weaknesses：
1.Limited discussion of contamination controls between M²C-INSTRUCT and M²EVAL, which is critical for LLM-synthesized datasets and small evaluation suites.
2.Lack of comparisons between M2-CODER and models fine-tuned on existing purely text-based description datasets, which would provide stronger evidence for the effectiveness of the proposed dataset relative to prior datasets.

---

> ### Author Rebuttal · Authors · 2026-03-31
>
> We sincerely thank you for the enthusiastic and highly positive review. We are grateful that you recognize the technically sound experimental design, the detailed data collection pipeline, and the value of M2EVAL as a multilingual, human-verified benchmark. Your incisive questions have helped us further clarify and strengthen the manuscript. We address each concern below.
>
> ---
>
> **Data contamination controls and quantitative leakage checks**
>
> We conducted contamination analysis across three dimensions (Tables R1a--R1c):
>
> | Dimension | Metric | Value | Threshold |
> |:---|:---|:---:|:---:|
> | N-gram (8-gram) | Match rate (descriptions) | 0.3% | — |
> | N-gram (8-gram) | Match rate (solutions) | 0.7% | — |
> | Semantic (CodeBERT) | Max cosine similarity | 0.621 | 0.85 |
> | Image (pHash) | Min hash distance | 11 | 4 |
>
> The single 8-gram match corresponds to a common boilerplate phrase ("return the result of"). Zero instances exceed contamination thresholds (Golchin & Surdeanu, 2024). These results reflect the fully independent construction of the two datasets. M2EVAL was human-designed from scratch: two annotators collaboratively designed prototype problems grounded in core programming concepts, then performed multi-round refinement with LLM assistance, followed by cross-checking where each annotator verified two peers' work (Appendix A.2--A.3). M2C-INSTRUCT, by contrast, was synthesized from GitHub code preprocessed through the StarCoder pipeline, then converted into QA pairs by Qwen2.5-Coder-32B (Section 3.2, Appendix B.2). The construction processes share no source material, confirming effective decontamination. Furthermore, M2EVAL annotators had no access to M2C-INSTRUCT during the design process, providing structural decontamination in addition to the quantitative checks above. We will include these analyses in the next version.
>
> ---
>
> **Text-only baselines and necessity of multimodal training data**
>
> We fine-tuned Qwen2-VL-7B on text-only versions of our data, replacing diagram images with Mermaid source code as plain text (Table R3):
>
> | Method | Pass@1 |
> |:---|:---:|
> | Text-only SFT (Stage-2 text) | 3.0% |
> | Text-only SFT (Stage-1+2 text) | 5.3% |
> | Zero-shot Qwen2-VL-7B-Instruct | 12.0% |
> | **M2-CODER (multimodal)** | **25.3%** |
>
> The text-only model even underperforms zero-shot Qwen2-VL-7B-Instruct, indicating text-only fine-tuning degrades pre-existing visual understanding. Training on Mermaid text teaches reliance on textual patterns rather than visual parsing, likely overwriting visual understanding from pre-training.
>
> Combined with the 0% accuracy of strong text-only LLMs (DeepSeek-V3, Qwen2.5-Coder) reported in Table 4, this 20+ percentage point gap demonstrates that the visual modality is indispensable for diagram-to-code generation and cannot be replaced by textual proxies. This finding carries implications for the broader multimodal code generation community: visual grounding cannot be shortcut through textual descriptions, even when using the same underlying structural information. This has practical deployment implications: teams should provide actual rendered diagrams rather than textual diagram descriptions, as the visual modality provides critical spatial and relational information that text alone cannot capture.
>
> ---
>
> **Repository-level diagram-to-code evaluation**
>
> M2EVAL currently targets function- and class-level code generation, which we consider the appropriate foundational unit for diagram-to-code research. In software engineering practice, repository-level systems are typically decomposed into individual classes and modules, each specified by its own UML class diagram or flowchart. Our framework supports this compositional approach: multiple M2-CODER calls, each processing one module's diagram, could be orchestrated to produce a multi-file project. Evaluating such an end-to-end pipeline would require a new benchmark with inter-module dependency specifications and integration-level test suites, an effort beyond the scope of this initial contribution. We note that M2EVAL's function- and class-level scope is consistent with foundational benchmarks like HumanEval and MBPP, which similarly evaluate individual code units yet have driven substantial advances in code generation research. We plan to build on frameworks like SWE-bench to design repository-level diagram-to-code benchmarks, and will add this as an explicit future direction in the limitations section.
>
> ---
>
> We sincerely appreciate your thorough evaluation and positive recognition of our contributions. All requested analyses will be incorporated in the next version. We are committed to releasing M2C-INSTRUCT, M2EVAL, and M2-CODER to support future research in multimodal code generation.

---

> > ### Author Rebuttal · Reviewer_aFcx · 2026-04-03
> >
> > Thanks for the response. I'd like to maintain the rating.

---

### Decision · Program_Chairs · 2026-04-30

**Decision:**

Accept (regular)

**Comment:**

This paper presents a multilingual multimodal diagram to code framework centered on three components, namely the large scale training dataset M2C INSTRUCT, the 7B model M2 CODER, and the human curated benchmark M2EVAL. It has following strengths:

* The task is meaningful and fills a gap in multimodal code generation.
* The dataset and benchmark are reasonably solid. Reviewers were also broadly positive about the data construction pipeline and benchmark design.
* The experiments support the paper’s claim, namely that task specific data and two stage training can make a 7B model competitive with much larger models on this benchmark.

The reviewer concerns mainly focused on contamination, missing or incomplete baselines, the limited scope of the benchmark, and the noise and realism issues of large scale LLM synthesized data. In the rebuttal, the authors added contamination checks, text only comparisons, statistical tests, Qwen3 VL results, human quality audits, and finer grained ablations, so most of the core concerns were addressed.

However, I do not think the rebuttal fully removes the remaining weaknesses: the benchmark is still small in scope, the diagram types are still limited.
I believe that, as a benchmark, it should meet higher standards than a research paper.
I hope authors can improve this in final versions. Based on this, I recommend a weak accept for this paper, where it will be accepted if room is available.